

# H-migration in peroxy radicals under atmospheric conditions

Luc Vereecken[1,2,3], Barbara Nozière[4]

[1] Department of Chemistry, Katholieke Universiteit Leuven, Celestijnenlaan 200F, 3000 Leuven, Belgium
[2] Max Planck Institute for Chemistry, Hahn-Meitner-Weg 1, 55128 Mainz, Germany
[3] Institute for Energy and Climate Research, IEK-8: Troposphere, Forschungszentrum Jülich GmbH, 52428 Jülich, Germany
4 IRCELYON, CNRS and Université Claude Bernard Lyon1, Avenue Albert Einstein 2, 69626 Villeurbanne, France

Correspondance to: Luc Vereecken (l.vereecken@fz-juelich.de)

**Abstract.** A large data set of rate coefficients for H-migration in peroxy radicals is presented, and supplemented with literature data to derive a structure-activity relationship (SAR), for the title reaction class. The SAR supports aliphatic $RO_2$ radicals, as well as unsaturated bonds and β-oxo substitution both endo- and exo-cyclic to the transition state ring, and α-oxo (aldehyde), –OH, –OOH, and –$ONO_2$ substitutions, including migration of O-based hydrogen atoms. Also discussed are –C(=O)OH and –OR substitutions. The SAR allows predictions of rate coefficients k(T) for a temperature range of 200 to 450 K, with migrations spans ranging from 1,4 to 1,9-H-shifts depending on the functionalities. The performance of the SAR reflects the uncertainty of the underlying data, reproducing the scarce experimental data on average to a factor of 2, and the wide range of theoretical data to a factor 10 to 100 depending also on the quality of the data. The SAR evaluation discusses the performance in multi-functionalized species. For aliphatic $RO_2$, we also present some experimental product identification that validates the expected mechanisms. The proposed SAR is a valuable tool for mechanism development, experimental design, and guides future theoretical work, which should allow rapid improvements of the SAR in the following years; relative multi-conformer TST (rel-MC-TST) kinetic theory is introduced as an aid for systematic kinetic studies.

## 1 Introduction

H-migration in alkylperoxy radicals, $RO_2$, is a well-known reaction in combustion processes, where it is a major path in OH radical generation in cool flames (Battin-Leclerc et al., 2011). Over the last few years, $RO_2$ H-migration is also studied intensively at ambient temperatures, owing to its role in important atmospheric processes, such as the (re)generation of OH and $HO_2$ radicals, and the formation of low-volatility oxygenated compounds and highly oxygenated molecules (HOMs) (Bianchi et al., 2019; Ehn et al., 2017; Hallquist et al., 2009; Jokinen et al., 2014; Vereecken et al., 2018). The wide range of implications of these reactions is due to the properties of the hydroperoxides formed in the H-migration (R1).





Firstly, RO$_2$ H-migrations can be chained, having multiple H-migration reaction that sequentially add O$_2$ moieties to a molecule (R2 and R3). Secondly, by H-bonding and inductive effects the –OOH group facilitates faster H-migration reactions, enhancing the formation of highly oxidized molecules. The bond strength of the α-OOH hydrogen atoms is

reduced, allowing for fast H-migration of these H-atoms (R4); the resulting α-OOH alkyl radicals were shown theoretically to be unstable, decomposing spontaneously to a carbonyl compound with an OH radical co-product that acts as an important oxidant in the atmosphere (Vereecken et al., 2004), and is responsible for chain branching in low-temperature combustion (Battin-Leclerc et al., 2011). Finallly, hydroperoxide-substituted RO$_2$ radicals can rapidly scramble the peroxy H-atoms (R5) (Jørgensen et al., 2016; Miyoshi, 2011; Nozière and Vereecken, 2019), allowing access to subsequent reactions and

products that may not be accessible from the originally formed HOOQO$_2$.

The RO$_2$ H-migration reactions have a comparatively high barrier and are often too slow to compete against the traditional atmospheric bimolecular reactions of peroxy at room temperature, i.e. their reaction with NO, HO$_2$, and other RO$_2$ radicals (Atkinson, 2007; Jenkin et al., 2019). These latter reactions have been studied extensively and are included in all chemical models, while their rate coefficients and products can be reasonably well predicted as function of the RO$_2$ molecular

structure (Jenkin et al., 2019). The rate coefficient of the unimolecular reactions, however, depends strongly on the span of the H-migration, the substitution pattern around the migrating H-atom and peroxy radical moiety, and the nature of the substitution. In particular, unsaturated oxygenated RO$_2$ radicals were found to have H-migration reactions that are competitive in pristine environments. RO$_2$ H-migration reactions are typically not well-represented in chemical atmospheric models, as they remain poorly characterized and no generally applicable quantitative structure-activity relationship (SAR) is

available to aid in the construction of mechanisms (Vereecken et al., 2018).

Though high-quality theoretical work has been available for RO$_2$ H-migration for quite some time, as discussed below, direct experimental data on the rate coefficients remains limited. Recent work by Nozière and Vereecken (2019) studied H-migration in aliphatic RO$_2$, by combined experimental and theoretical methodologies, examining migration of primary,



secondary and tertiary H-atoms over spans from 1,5-H- to 1,8-H-migrations. This experimental study of aliphatic $RO_2$ H-migration is currently the only available direct measurement of aliphatic $RO_2$ H-migration rate coefficients, and thus represent important reference data in the development of SARs. H-migration from aliphatic carbons is a critical step in the formation of highly oxygenated molecules where oxygen-to-carbon ratios O:C as high as 2 have been observed, implying that (nearly) all carbons have been oxygenated (Bianchi et al., 2019; Ehn et al., 2017). All other experimental studies we are aware of involve (unsaturated) oxygenated peroxy radicals, mostly $RO_2$ radicals derived from isoprene and its oxidation products (Crounse et al., 2011, 2012, 2013; Jorand et al., 2003; Perrin et al., 1998; Praske et al., 2018, 2019; Xu et al., 2019). Modern explicit atmospheric models have extensive representations of this isoprene $RO_2$ chemistry (Carter, 2010; Jenkin et al., 2015; Novelli et al., 2019; Wennberg et al., 2018), derived mostly from the Leuven Isoprene Mechanisms LIM0 and LIM1 (Peeters et al., 2009, 2014) .

A recent review of $RO_2$ chemistry by Jenkin et al. (2019), aimed at developing SARs for use in mechanism development, included a short list of recommendations for H-migrations but although the authors rely on theoretical data they refrained from developing a generally applicable SAR based on the systematic theoretical studies available in the literature. More recently, Nozière and Vereecken (2019) proposed a simple SAR for aliphatic and OOH-substituted $RO_2$ radicals, combining their theoretical and experimental data with the systematic theoretical studies by Miyoshi (2011), Sharma et al. (2010), and Otkjær et al. (2018). Older work by Vereecken and Peeters (Vereecken, 2012; Vereecken and Peeters, 2010b) presented an extensive SAR covering aliphatic, unsaturated, and oxygenated $RO_2$, but this data was never formally published. Extensive theoretical data is also available on multi-functionalized $RO_2$ radicals (e.g. Fuchs et al., 2018; Jørgensen et al., 2016; Møller et al., 2019; Novelli et al., 2019; Praske et al., 2018, 2019; da Silva, 2010b, 2010a, 2011). Combined, this theory-based data set is sufficiently large to derive a first-order SAR covering many of the needs in atmospheric chemistry, and serving as a basis for future improvements. If the rate coefficient predictions carry an uncertainty of an order of magnitude or less, many questions regarding the competitiveness of the $RO_2$ H-migration relative to other $RO_2$ loss processes could be answered, and (semi-)quantitative models could be constructed.

In this work, we present an additional large theory-based data set, and derive a first-order SAR for $RO_2$ H-migration covering a wide range of chemical functionalities, with a reasonable level of reliability of ~1 order of magnitude, depending on the molecular structure. A full review of the $RO_2$ literature is outside the scope of the current work, and only data used in deriving or validating our SAR is discussed. Work is ongoing to systematically re-evaluate the data at higher levels of theory or fill in missing data, and the current SAR helps to identify the most relevant reactions on which to dedicate limited computational resources. Where possible, experimental data will be used to validate and/or fine-tune the rate coefficient predictions. Unfortunately, not all experimental data is readily incorporated. For example: at high temperatures, the rate of reaction is determined strongly by the entropic changes from the reactant chain internal rotors to the cyclic transition states, while at lower temperatures the energy barrier height and tunneling of the H-atom are the most critical parameters. This transition leads to strongly non-linear Arrhenius behavior, making extrapolation of the high temperatures data to ambient conditions more difficult. Similarly, the experimental data often provides bulk rate coefficients, convoluting multiple elementary reactions that may act on different time scales. One should bear in mind that the rate coefficient for H-migration may not be sufficient to determine the impact of a reaction. Specifically, the reverse reaction must be slow enough to allow



the (alkyl) reaction product to undergo a subsequent transition or chemical reaction rather than return to the parent $RO_2$ radical. Examples of such subsequent capture reactions are $O_2$ addition on the alkyl radical site (in the atmosphere typically at a rate of $k_{O2} \sim 10^6$ s$^{-1}$ (Atkinson, 2007)) or fast decomposition of the product. The reverse H-migration is likewise slowed down by physicochemical effects such as allyl or vinoxy resonance stabilization of the radical electron, hydrogen bonding of the –OOH product moiety, etc. Several authors have listed rate coefficients for reverse reactions (e.g. Jørgensen et al.,

2016; Miyoshi, 2011; Otkjær et al., 2018); by and large, faster $RO_2$ H-migration reactions tend to have slower reverse reactions, such that in atmospheric conditions the reverse reactions are typically relevant only for those $RO_2$ reactions already too slow to compete. The experimental data on $RO_2$ H-migration reactions is not only scarce, they are often also indirect and obtained for molecules with complex (oxygenated) substitution, with rate coefficients inferred from product observations without experimental confirmation that the assumed underlying mechanisms are correct and complete. A

recent paper by Novelli et al. (2019) for example briefly discusses how isoprene mechanisms have changed responding to new data. The recent experiments described in Nozière and Vereecken (2019) are the most direct measurements currently available of $RO_2$ H-migration reactions. In this work, we supplement these experiments with an analysis of the stable products in these studies, validating the expected reaction mechanisms.

## 2 Methodology

### 2.1 Quantum chemical calculations

The supporting information lists a large data set for $RO_2$ H-migration reactions, as obtained over the last decade by our research group. Many levels of theory were employed for geometry optimization, ranging from older B3LYP/6-31G(d,p) to modern M06-2X-D3/aug-cc-pVTZ methodologies, and combined with single point energy calculations ranging from CBS-QB3, to CCSD(T) calculations with extrapolations to the complete basis set limit. The supporting information lists the

methodologies. Each methodology has its own advantages and drawbacks, and unfortunately its own specific uncertainties and systematic bias. To reduce the method-induced scatter in the data set, we preferentially limit ourselves to data obtained at three levels of theory. The first is the CCSD(T)/aug-cc-pVTZ//M06-2X/aug-cc-pVTZ level of theory (wavenumber scaling factor 0.971) (Bao et al., 2017; Dunning, 1989; Purvis and Bartlett, 1982; Zhao and Truhlar, 2008), a high-end methodology that has been used extensively by our group in recent publications (Fuchs et al., 2018; Novelli et al., 2019;

Nozière and Vereecken, 2019), and was found by Nozière and Vereecken (2019) to reproduce the direct experimental data for $RO_2$ H-migration faithfully. A second set of calculations are based on CBS-QB3//B3LYP/6-31G(d,p) (5 d-orbitals, scaling factor 0.977) (Bao et al., 2017; Becke, 1992; Lee et al., 1988; Montgomery et al., 2000). These calculations carry a somewhat larger error and scatter, but we find the obtained rate coefficients to mostly remain within a factor of 5 compared to more rigorous methodologies. A third set of data is based on CCSD(T)//M05-2X/6-311G(d,p) calculations (scaling

factor 0.964) (Bao et al., 2017; Purvis and Bartlett, 1982; Zhao et al., 2006), of a reliability intermediate between the two earlier methods. Finally, relative rate estimates are based directly on B3LYP/6-31G(d,p) calculations; these relative rates





estimates (see below) build solely on energy and entropic changes, not their absolute value, benefiting from cancellation of error.

The supporting information provides some more details on the calculation, including additional performance tests. It was
found that CCSD(T) extrapolation to the complete basis set is not cost-effective at this time, considering the other contributions to the uncertainty of the SAR. IRCMax calculations show that the TS geometries obtained using B3LYP are subject to larger additional errors, sometimes exceeding 1 kcal mol$^{-1}$ on the absolute barrier height, compared to using more modern functionals such as M05-2X and M06-2X.

## 2.2 Absolute rate coefficient calculations

The rate coefficients of all reactions are calculated between 200K and 450K using multi-conformer transition state theory (MC-TST) as described by Vereecken et al. (Vereecken and Peeters, 2003), and applied extensively and successfully in earlier work. The method is based on canonical transition state theory (CTST) calculations (Truhlar et al., 1996) in a rigid rotor harmonic oscillator approximation but explicitly including all conformers of reactants and transition states to accommodate the effect of internal rotation, and the differences in energy and entropy of each of the conformers. The
calculations are performed at the high-pressure limit; earlier work has found that the rates of RO$_2$ H-migration are sufficiently slow to be negligibly affected by pressure fall-off at 1 atm (Miyoshi, 2012; Peeters et al., 2014; Xing et al., 2018; Zhang and Dibble, 2011). Tunneling is incorporated using asymmetric Eckart tunneling (Eckart, 1930; Johnston and Heicklen, 1962); for most H-migrations the reaction energy is estimated from an explicit product characterization (usually 1 conformer only), while for the remaining reactions the product energy is assumed to be similar to analogous reactions with
similar product characteristics.

## 2.3 Relative rate coefficient calculations

Relative rate coefficient estimates were performed using the relative multi-conformer transition state theory (rel-MC-TST) described in the supporting information. Briefly, this method derives a rate coefficient from an MC-TST prediction for a reference compound, and the change in rate induced by a modification of the molecular structure from the reference
compound to the target compound, quantified by the characterization of a (small) subset of reactant and transition state conformers of the reference and target reaction. The method provides a cost-effective estimate of the rate coefficient, where the approximation becomes better as the molecular structure between reference and target reaction is more similar, and as more conformers are included in the relative rate change calculations. When including all conformers the method becomes the equivalent of a full MC-TST calculation. We have performed a set of calculations using heptane and octane as reference
compounds, looking at the impact of methyl-, hydroxy-, and oxo-substitution, as well as the impact of double bond. This new methodology will be benchmarked in a separate paper.

## 2.4 Theoretical literature data

A few extensive theory-based data sets have been published over the last decade which, combined with the data set presented in this work and our earlier publications, form the basis of our current work. Several of these studies were aimed





at combustion temperatures, and don't include data below 300K. As a result, these rate equation are unlikely to properly describe the tunneling-induced curvature in the Arrhenius plot below 300K, possibly underestimating the rate coefficients at the lowest temperatures considered in this work, 200K. We briefly list some methodological aspects of a selection of papers. Sharma et al. (2010) studied a systematic series of H-migrations, using very high levels of methodology, explicitly accounting for quantum internal rotation but using less accurate Wigner tunneling corrections, at temperatures ranging from

300 to 1500K. This work contains aliphatic $RO_2$ compounds, examining 1,3- to 1,7-H-shifts looking at abstraction of primary, secondary and tertiary H-atoms. A study by Miyoshi (2011) used similarly high levels of theory, with explicit consideration of internal rotation and applying more accurate Eckart tunneling corrections, at temperatures 500-1250 K and pressures from 0.1 to 100 atm. This work examines aliphatic and hydroperoxy-substituted $RO_2$, examining 1,4- to 1,8-H-migrations and categorizing the reactions based on substitution on the carbons bearing the –OO$^{\bullet}$ radical and the migrating

H-atom; this categorization is adopted in the current work. The Sharma et al. (2010) and Miyoshi (2011) data sets form a large part of the current SAR, and are considered to be generally reliable. Nozière and Vereecken (2019) recently derived a simple SAR based on these data sets, finding excellent agreement with experimental measurements. Data by Zang and Dibble (2011) covers a smaller set of migrations, mostly limited to 1,5-H-migrations, but includes a set of unsaturated hydrocarbons; the rate equations are derived for temperatures 300-2500K. Otkjær et al. (2018) recently performed a

systematic study of 1,4- through 1,10-H-migrations, examining a wide range of substitutions (alkyl, oxo, F-atom, hydroxy, hydroperoxide, nitrate, methoxy, and unsaturated functionalities) at high levels of theory, with a MC-TST-based kinetic analysis. These authors examined the 290-320K temperature range, but due to the chosen k(T) function the temperature dependence can not be reliably extrapolated beyond this range. Mohamed et al. (2018b, 2018a) performed a systematic study on hydroperoxide-substituted $RO_2$ radicals in the temperature range 300-1500K. The internal rotors are represented by

a Pitzer and Gwinn approximation based on averaged barriers to rotation in the lowest-energy conformer, as obtained by explicit (partial) optimization of the rotational energy profile. At this time, we choose not to incorporate the results by Yao et al. (2017); their predictions for the 500-1200 K range are not compatible with the experimental data, overestimating the room temperature rate coefficients by several orders of magnitude.

   In addition to these larger data sets, we include data from a number of other publications, which are cited in the relevant

sections.

## 2.5 Theory-based rate coefficients

   The theoretical predictions for ~150 rate coefficient derived using MC-TST and rel-MC-TST rate theory are tabulated in the supporting information sects. 2 and 3. This data set collects rate coefficients obtained by our research group over the last decade, including some which have been presented before (Fuchs et al., 2018; Novelli et al., 2019; Vereecken, 2012;

Vereecken and Peeters, 2010b). The rate coefficients published by other authors, representing a few hundred data points, are not replicated here, but can be found in the respective publications.



### 2.6 Structure-Activity Relationship design

Merging the different data sets into a structure-activity relationship follows a set of steps for each rate coefficient in the data set. First, each reaction is categorized in one of the categories based on the migration span, and the substitution pattern around the migrating H-atom and the –OO$^\bullet$ moiety. Next, the rate coefficient is calculated across the temperature range 200-450K, or the temperature range where the kinetic expression is expected to be valid (e.g. 290-330K for the data from Otkjær et al. (2018)) to avoid introducing extrapolation errors. The k(298K) target rate coefficient at 298K is then obtained by geometrically averaging (equal weight) selected rate coefficients at that temperature. Source rate coefficients are within a factor 2 to 3 of the obtained average for aliphatic $RO_2$, indicating good agreement of the data sets; for other functionalities there can be significantly more scatter. To determine the temperature profile of the SAR rate coefficient, the rate coefficients over the 200-450 K temperature range are geometrically averaged (equal weight). Any data that is deemed of insufficient quality is excluded in the previous steps; the selected data sets, however, contain few excluded data. The most common reason for rejection is suspicion of low-temperature rate under-prediction due to limitations in the tunneling corrections and the extrapolation from higher temperatures. The subset of rate predictions in the k(T) profile prediction can be smaller than for determining k(298K), as not all data sets are valid over the required temperature range, or have limitations in their tunneling corrections and extrapolation from higher temperatures. The obtained k(T) temperature-dependent profile is then shifted to match the target k(298K) by altering the reaction barrier height, i.e. a correction term exp(Corr/T) is factored into the k(T) profile. Finally, the shifted k(T) rate coefficients are then fitted to a modified Arrhenius expression (Kooij expression) $k(T) = A \times (T/K)^n \times \exp(-E_a/T)$ and listed in the SAR tables below. Figure 1 shows an example of this procedure, showing the original literature data, and the final k(T) SAR rate coefficient. Figure 2 shows an example of one of the SAR classes where many data points are available, showing the scatter on the literature data, and the averaging effect of the SAR procedure. The supporting information contains a spreadsheet that has values and figures for all the underlying data and the derived SAR k(T).

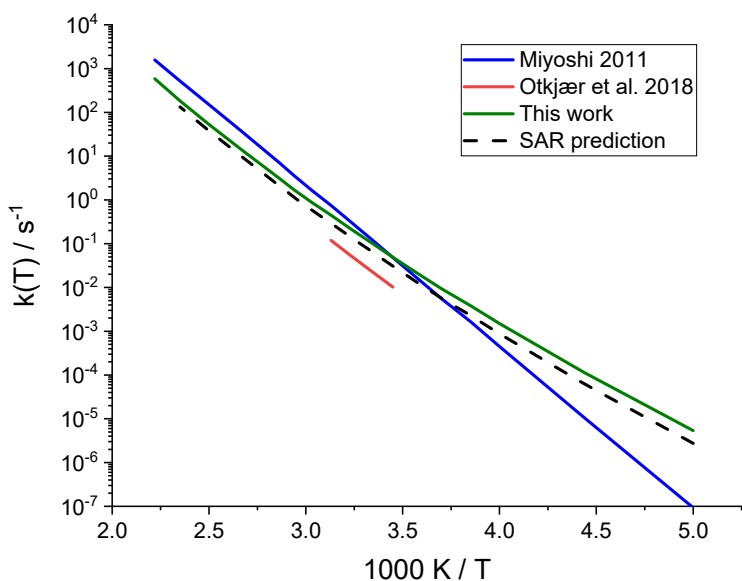

*Figure 1: Rate coefficient for 1,5-H-migration in an aliphatic alkyl peroxy radical with a >CH(OO•) radical group, and a tertiary (–CH<) migrating H-atom. The SAR derivation matches a geometric average at 298 K, and uses the most reliable temperature-dependence accounting for tunneling at low temperatures.*






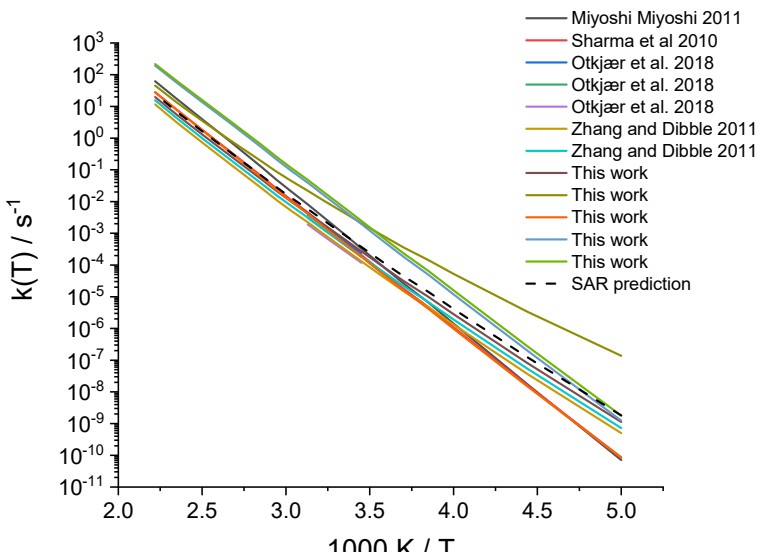

*Figure 2: Rate coefficient for 1,5-H-migration in an aliphatic alkyl peroxy radical with a $>CH(OO^\bullet)$ radical group, and a secondary ($-CH_2-$) migrating H-atom. The literature data for the various reactions in this class are scattered over a factor 22; the final SAR predictions is within a factor 7 of all data at 298 K, and incorporates the average temperature dependence of the 5 predictions that explicitly treat tunneling corrections below 300 K.*

The resulting k(T) expressions are expected to be less sensitive to the idiosyncrasies of the individual methodologies than any specific theoretical methodology —at least where multiple data sets are available— and to provide a good prediction of the class-specific behavior of $RO_2$ H-migration, especially when multiple molecules are included within the class thus averaging out the molecular specifics. The use of geometric averaging, and temperature-dependent correction factors reflect that the largest contributions to the uncertainty of theoretical predictions are due to uncertainties on the barrier height and

tunneling, both of which affect the rate coefficient non-linearly as a function of temperature. The transfer to the final SAR expression of temperature-dependencies sourced from the most reliable k(T) predictions ensures the best possible temperature dependence, and avoids extrapolation problems from high-temperature data.





In some cases, there is insufficient data to derive a temperature dependence, even though room temperature data is available. In such cases, given that the temperature dependence is comparable in all $RO_2$ H-migration after correction by a
shifting factor exp(Corr/T), we resort to using the temperature dependence of a similar reaction, the average T-dependence in the reaction class, or the average T-dependence of a comparable reaction class, to derive a rate expression. Finally, in case no data at room temperature is available for a SAR class with a particular substitution, we base a tentative rate expression on the rate difference at 298K between the corresponding classes for aliphatic $RO_2$. The uncertainty on these latter rate expressions is significantly larger than classes for which direct data is available, and is likely no better than 2
orders of magnitude when factoring in the uncertainties on the underlying data and on transferability of kinetic properties. We provide such rate coefficients mostly in an attempt to provide a rough estimate, rather than providing no guidance at all. The different levels of approximations on the rate coefficient expressions are marked in the respective tables below.

The SAR parameters are presented as a set of lookup tables, showing both the k(298 K) value as the parameters for the modified Arrhenius rate expression $k(T) = A \times (T/K)^n \times \exp(-E_a/T)$, combined with a few correction terms that only adjust
the effective activation energy. An earlier H-migration SAR for alkoxy radicals (Vereecken and Peeters, 2010a) presented the results as distinct contributing factors for ring strain, rigidity, conformers, tunneling, etc.. Other authors employed expressions factoring in a rate equation and a tunneling correction ($\sim\exp(A/T^3)$). These approaches are thought to be cumbersome in practice due to incompatibilities with existing tool chains, and we feel the current approach is significantly more accessible both for direct lookup by humans, as for implementation in model generation software.

At this time, we choose not to focus the SAR on multi-functionalized compounds, i.e. we derive our SAR for aliphatic and mono-functionalized compounds, in as much as data is available for these. We also do not distinguish between stereo-isomers of chiral carbons (R/S-geometries) even though we find that the rate coefficients can differs by up to two orders of magnitude for complex, multi-functionalized species with multiple chiral carbons. As discussed below in more detail, we only consider H-migrations in *cis-* and *gem*-substitutions in unsaturated species.

While not included in the derivation of the SAR, the data on multi-functionalized molecules is used to determine the performance of the SAR when extrapolated to such more complex molecules. The scope of applicability for spectator functionalities, H-migration affected by multiple functionalities, and cyclic compounds, is examined in a later section.

### 2.7 Experimental methodology

For the experimental studies of the H-migration reaction of aliphatic $RO_2$, the methodology has been described in detail in
earlier work (Nozière and Vereecken, 2019); we only provide a brief description here.

### 2.7.1 Radical generation

H-migration reactions for a series of aliphatic $RO_2$ (1-butyl$O_2$, 1-pentyl$O_2$, 1-hexyl$O_2$, ethyl-hexyl$O_2$ and dimethyl-hexyl$O_2$) were studied experimentally at room temperature (T = 298 ± 5 K) in a cylindrical flow reactor (Quartz, inner diameter 5 cm, length 120 cm). The $RO_2$ were produced in a flow of synthetic air (3 – 4 sLm, standard temperature = 273 K and pressure = 
1 atm), to which was added an organic precursor $3 \times 10^{15}$ to $5 \times 10^{16}$ molecule cm$^{-3}$. For 1-butyl$O_2$, 1-pentyl$O_2$, 1-hexyl$O_2$, and ethyl-hexyl$O_2$, the precursor was the iodinated analog: iodobutane, iodopentane, iodohexane, and 2-ethylhexyl iodide,





respectively. For dimethyl-hexylO$_2$ it was 2,5-dimethylhexane. The radicals were produced by irradiating the flow mixture over 30 – 40 cm at mid-length of the reactor with 4 fluorescent lights (Philips TL12, 40 W, emitting over 280-400 nm). The iodinated precursors were photolyzed directly, thus ensuring the formation of a single radical in each case. For instance, for

1-butylO$_2$:

$$I\text{-}C_4H_9 \rightarrow C_4H_9 + I \tag{R6}$$

$$\cdot C_4H_9 + O_2 + M \rightarrow C_4H_9O_2 + M \tag{R7}$$

For dimethyl-hexylO$_2$, Cl$_2$ (~ $10^{15}$ molecule cm$^{-3}$) was also added to the reactor and photolyzed in the presence of dimethyl hexane:

$$Cl_2 + h\nu \rightarrow 2\ Cl \tag{R8}$$

$$Cl + C_8H_{18} \rightarrow \cdot C_8H_{17} + HCl \tag{R9}$$

$$\cdot C_8H_{17} + O_2 + M \rightarrow C_8H_{17}O_2 + M \tag{R10}$$

### 2.7.2 Radical and stable product detection

A small fraction (50 – 100 sccm) of the reaction mixture was sampled through a sampling line at the output of the flow

reactor and analyzed with a quadrupole Chemical Ionization Mass Spectrometer (CIMS) using proton transfer at high pressure as ionizing technique (Hanson et al., 2004, 2003; Nozière and Hanson, 2017; Nozière and Vereecken, 2019). With this technique, the RO$_2$ and their stable products were detected by reacting with proton-water clusters, H$^+$(H$_2$O)$_n$ with n = 1 to 5, in the ionization region of the CIMS:

$$RH + H^+(H^2O)_n \rightarrow RH(H_2O)_{n-m}H^+ + (H_2O)_m \tag{R11}$$

Thus, a compound of molecular weight M was typically detected at the ion masses M+1, M+19, M+37 and M+55. Typical resolution for the instrument was M/ΔM ~ 250 and the detection limit between $10^8$ and $10^9$ molecule cm$^{-3}$. The radicals were unambiguously distinguished from potential stable isomers in the mass spectra by adding NO periodically in the reactor, resulting in their consumption. The periodic variation of other masses upon addition of NO also confirmed the identification of the stable compounds of the photochemical system, and allowed to distinguish them from potential artefacts or pollution

in the flow system.

## 3 Structure activity relationship

### 3.1 Aliphatic RO$_2$

Linear and branched aliphatic peroxy radicals are described by the SAR rate coefficients in Table 1, and is derived based on our calculations (see supporting information) and selected literature data (Miyoshi, 2011; Otkjær et al., 2018; Praske et al.,

2019; Sharma et al., 2010; Zhang and Dibble, 2011). No rate coefficients are included for 1,3-H-migration, as these have been found to have high barriers and very low rate coefficients, k(298K) ≤ $10^{-16}$ s$^{-1}$, making them irrelevant for atmospheric chemistry (Sharma et al., 2010). 1,4-H-migrations have likewise high barriers and low rate coefficients, k(298K) ≤ $10^{-7}$ s$^{-1}$, for aliphatic compounds and are negligible in atmospheric conditions. Furthermore, the reaction barriers for 1,4-H-



migration tend to be higher than for the (also slow) $HO_2$ elimination forming an alkene, making H-migration a minor
channel even if accessible. The 1,4-H-migration is included here mostly to serve as a basis for substituted 1,4-H-migration,
such as aldehydic H-migration where significantly lower barriers are found. Rate coefficients for 1,8-H-migration are
similarly low, $k(298K) \leq 10^{-6}$ s$^{-1}$, due to the entropic disadvantage of losing a large number of internal rotors in the cyclic
transition state; again, we include them here mostly for possible use as a reference rate coefficient. H-migration at even
larger spans, e.g. 1,9- or 1,10-H-shifts, have been studied by Otkjær et al. (2018), finding even lower rate coefficients and
negligible contribution of these long-range migrations. Aliphatic $RO_2$ H-migration is then mostly governed by 1,5-H-, 1,6-
H-, and 1,7-H-migrations. The predicted rate coefficients in Table 1 show that substitution around the migrating H-atom is
of critical importance, with tertiary H-atoms migrating several orders of magnitude faster than primary H-atoms.
Substitution around the $–OO^\bullet$ moiety has less impact on the rate, and the impact varies with the substitution. We find no
significant influence of the length of the alkyl chain attached, in agreement with e.g. Otkjaer et al. (2018).

## 3.2 Unsaturated $RO_2$

To parameterize the impact of double bonds in the peroxy radicals, one should in principle account not only for the
substitution around the peroxy and migrating H-atom, but also whether the double bond has zero, one or both carbon atoms
within the TS cycle, whether the product radical is stabilized by allyl resonance, and if so, what the substitution is on the
second radical site, and possibly on the center carbon of the $C^\bullet–C=C$ allylic group. Abstraction of vinyl H-atoms, $=CH–$, is
known to be negligibly slow at room temperature, of the order of $k(298K) \sim 10^{-10}$ s$^{-1}$ (Zhang and Dibble, 2011), and is not
considered in this work. There is little direct information on substitution on the central carbon of the allylic radical group,
but the scarce data suggests the impact on $k(298K)$ to be of the order of a factor of ~2 for H-atom versus alkyl substitution.
This comparatively small factor is not separable from the uncertainty in our data set, and our SAR reaction classes don't
distinguish between these substitutions. Although it is technically feasible to migrate between trans-substituents, we find
significant barriers for such processes even when forming allyl-stabilized alkyl radicals, e.g. 46 kcal mol$^{-1}$ for 1,6-H-
migration in $E$-2-butene-1-peroxy radicals, or 31 kcal mol$^{-1}$ for 1,7-H-migration in $E$-3-pentene-1-peroxy radicals. To
reduce the strain in the transition states very large migrations spans would then be required, leading again to low rate
coefficients due to entropic disadvantage. Hence, in this work we focus only on substituents located on the same side of the
double bond, i.e. gemini-substitution on the same carbon in the double bond, or cis-substituents on the same side of the
double bond carbon atoms.

As already shown extensively, migration of an allylic H-atom is found to be significantly faster than migration of an
aliphatic carbon (Otkjær et al., 2018; Zhang and Dibble, 2011). The scatter between the available data sets is significantly
larger than for aliphatic $RO_2$, with differences of over an order of magnitude between methodologies and/or authors. The
cases with the largest deviation from the average can often be traced to the absence of important (lower-energy) conformers
in the multi-conformer analysis. In some compounds, we find large differences between the rate coefficients for endo- or
exocyclic double bonds in the TS ring, up to two orders of magnitude on $k(298K)$, though for other H-migrations the
difference is negligible, making it hard to judge whether this is due to methodological scatter, or to the impact of ring strain



and internal rotations on the entropy. At this time, we choose to distinguish between endo- and exo-cyclic double bonds until more data is available.

We have only a single data point, a 1,7-H-migration in *Z*-2-heptene-1-peroxy, where a double bond is present in the TS ring, but where the migrating hydrogen is not an allylic H-atom; the rate coefficient is indistinguishable from its aliphatic equivalent. As such, we only consider SAR classes for migration of allylic H-atoms, until more information is available on the entropic effect of a non-allylic double bond in the TS ring.

### 3.2.1 Allylic H-atom migration with a TS endocyclic double bond

The SAR expressions (see Table 2) for endo-cyclic double bonds in the TS ring are derived based on our calculations (see supporting information) and the data by Zhang and Dibble (2011).

### 3.2.2 Allylic H-atom migration with a TS exocyclic double bond

The SAR expressions (see Table 3) for endo-cyclic double bonds in the TS ring are derived based on our calculations (see supporting information) and selected literature data (Otkjær et al., 2018; Zhang and Dibble, 2011).

### 335 3.2.3 Allylic H-migration with a endo-cyclic *gem*-substituted double bond

The delocalizing double bond of the allylic H-atom can have only a single carbon in the TS cycle, which occurs when the $^\bullet$OO-bearing and the migrating-H-bearing substituent are in gemini-substitution (same carbon atom) of the double bond. This configuration is somewhat different than endo-cyclic double bonds with both carbons in the TS ring, due to differences in the number of internal rotors, rigidity, etc. *A priori*, one would expect these H-migrations to be entropically

disadvantaged comapred to endo-cyclic double bonds, due to the loss of a higher number of internal rotors in the *gem*-TS ring, and therefore slower than exo-cyclic double bonds. We have very few data points for such structures, mostly for methyl-H-migrations in isoprene-derived peroxy radicals after reaction with OH or $NO_3$, but the scarce data points do not indicate a significant difference between *gem*- and *endo*-double bonds, with rate coefficient differences of an order of magnitude at most. Until more systematic data is available, we recommend using the same rate coefficients for *gem*-

substitution as for *endo*-cyclic double bonds.

### 3.2.4 Triple and conjugated unsaturated $RO_2$

Conjugated, –C=C=C–, or triple, –C≡C–, unsaturated groups are characterized by a rigid structure with at least 3 to 4 atoms linearly aligned. H-migrations across such subgroups, i.e. with the unsaturation within the TS cycle, are geometrically very difficult, and only long-range migrations, 1,8-H-migration or further, might be feasible. Rate coefficients for these spans

tend to be rather low, and at this moment we assume H-migrations across conjugated or triple bonds to be negligible. There is not enough information to reliably derive rate coefficients for these unsaturations when outside the TS cycle and leading to resonance-stabilized alkyl radicals; at this time, we propose to apply the SAR for allylic H-atom migration with exocyclic double bonds.



### 3.3 Oxo-substituted RO$_2$

**3.3.1 Migration of aldehydic H-atoms**

The abstraction of aldehydic H-atoms is comparatively fast, k(298K) ≥ 0.5 s$^{-1}$, even for the short span of 1,4-H-migrations. Our calculations are the only data points in our data sets for mono-substituted aldehydic RO$_2$ radicals, and we resort to using information on hydroxy-substituted aldehydic RO$_2$ for some of the aldehydic SAR classes. This includes acrolein and methacrolein, as found in our data set, the theoretical data by Møller et al. (2016) and da Silva (2011), and the MACR

experimental rate determination by Crounse et al. (2012). As discussed in the section for OH-substituted RO$_2$, the β-OH substitution of the peroxy radical moiety could affect the rate coefficient, and we should allow for an overestimation of a factor of 10 in our SAR predictions for 1,4-H-migration. The OH-substitution is likely also the cause for the low rate of 1,5-aldehyde H-migration in 3-hydroxypropanal-3-peroxy described Møller et al. (2016). Additional calculations on aldehydic compounds would be valuable. The SAR parameters are listed in Table 4.

**3.3.2 Migration of β-oxo H-atoms**

The data set of Otkjær et al. (2018) combined with our data set allows an approximate evaluation of the impact of β-oxo substitution of the migrating H-atom, forming vinoxy-stabilized product radicals. The effect of this substitution has both an energetic component, due to the stabilization of the product radical, as well as an entropic component, where the carbonyl bond changes ring strain and rigidity. One should thus consider ring size as well as endo- or exo-cyclic position of the

>C=O moiety in the TS. Generally, we find that β-oxo substitution enhances the H-migration rates compared to analogous aliphatic RO$_2$. Exocyclic carbonyl bonds strongly enhance H-migration rates with strained rings, have a more limited effect with rings with less strain and geometric constraints, and are expected to have no significant effect with very large rings. This is in agreement with e.g. VOC H-abstraction rates by OH radicals, where a carbonyl group has little to no impact on the rate coefficient (e.g. ethane+OH versus acetone+OH). For endocyclic carbonyls, we find the opposite effect, where the

presence of a rigid carbonyl group in an already strained ring decreases the rate coefficient, but where for large rings it has no strong effect. We summarized the scarce data as a set of correction factors (see Table 5) on the rate coefficient of the corresponding aliphatic RO$_2$, as a function of the position of the carbonyl group and the migration span. We have no information available for 1,8-H-migrations.

**3.3.3 Migration in other oxo-substituted RO$_2$ radicals**

We have almost no data for peroxy radicals where the carbonyl is not in β-position to the migrating H-atom. For aliphatic H-migration in acylperoxy radicals, Møller et al. (2019) calculated rate coefficients in multi-substituted RO$_2$, finding faster reactions compared to aliphatic peroxy radicals. We tentatively summarize the differences as a factor exp(900 K/T), listed in Table 5. Knap and Jørgensen (2017) calculated an α-OOH 1,6-H-migration in an acylperoxy radical, finding a rate coefficient of 98 s$^{-1}$, a factor 4×10$^3$ faster than the α-OOH 1,6-H-migrations in aliphatic peroxy radicals; they also

characterized aldehyde-H-migrations by acylperoxy radicals that are significantly faster than in regular RO$_2$. Knap and Jørgensen (2017) also describe H-scrambling in acyl peroxyradicals, finding rate coefficients ranging from 10$^3$ to 10$^6$ s$^{-1}$ in



agreement or faster than the H-scrambling data described elsewhere in this work. At the same time, these authors find much slower reactions for the reverse H-scrambling from peracid groups (–C(=O)OOH). The above data indicates strongly that acylperoxy radicals are significantly more reactive towards H-migration owing to the higher stability of the product peracid group, and should be investigated in more detail.

We have no systematic information for endocyclic oxo substituents that are in β-position to neither the migrating H-atom nor the peroxy radical group. Our data set includes a single data point for an endo-β-oxo-substituted –OO$^\bullet$ group; our relative MC-TST calculation on this latter 1,6-H-migration indicates an impact of less than a factor of 5, but this value is not reliable enough to use as a basis for a SAR proposal. da Silva (2010a) calculated a rate coefficient $k(T) = 5.86\times10^3$ $T^{2.662}$ $\exp(-6358K/T)$ s$^{-1}$ for the 1,4-H-shift in $^\bullet$OOC(=O)CH=O, i.e. $k(298K) = 12$ s$^{-1}$, faster than the 1,4 aldehyde-H-migrations rate coefficients in the SAR. Due to the paucity of data, we currently make no recommendations for any of these types of RO$_2$ H-migrations with endo-cyclic oxo substitutions.

### 3.4 Hydroxy-substituted RO$_2$

#### 3.4.1 Migration of α-OH H-atoms

α-OH substitution is known to reduce the C–H bond strength compared to an alkyl substitution, allowing for faster H-abstraction (IUPAC Subcommittee on Atmospheric Chemical Kinetic Data Evaluation, 2017) or H-migration (Vereecken and Peeters, 2010a). In RO$_2$ radicals, we observe the same effect, with an increase in rate coefficient of about a factor 10 at room temperature. The SAR predictions for migration of α-OH-H-atoms are listed in Table 6, as derived from our calculations (see supporting information) and selected literature data (Jørgensen et al., 2016; Otkjær et al., 2018; Praske et al., 2018, 2019). There is direct experimental data available at higher temperatures, ~450-525K (Jorand et al., 2003; Perrin et al., 1998). Our theoretical calculations reproduce these experimental results well, within a factor ~5, and the SAR rate equations reproduce these values within a factor 3.5, well within our SAR accuracy goal.

There is very little information for gemini-di-hydroxy substitution; we are only aware of our own calculations on the 1,3- and 1,4-H-migration in hydroxy-methylperoxy versus dihydroxy-methylperoxy (see supporting information). These data do not provide a good reference for developing a generally applicable SAR correction, as the product radical decomposes, forming OH (1,3-H-shift) or HO$_2$ (1,4-H-shift), respectively. The scarce data does suggest that adding a second –OH substitution increases the H-migration rate significantly.

#### 3.4.2 Migration of β-OH H-atoms

The scarce data available on non-multi-functionalized RO$_2$ with β-OH substitution next to the migrating H-atom (this work, Praske et al., 2018, 2019; da Silva and Bozzelli, 2009) suggests that β-OH substitution next to the migrating H-atom slows down H-migration, on average by a factor 0.17 at room temperature, both when the OH group is in endo- or exo-position on the TS ring. This corresponds to an increase in barrier height of about 1 kcal mol$^{-1}$. The reason appears to be that the H-bonding is absent or less favorable in the TS cycle due to geometric constraints, as opposed to the reactant where the peroxy



radical group can interact freely with the –OH group. We propose that the rate coefficients for H-migration of aliphatic H-atoms with a β-OH substituent are corrected by a factor exp(-528K/T) to account for this effect (see Table 5).

The information of β-OH substition of the –OO• group is even more scarce (Jørgensen et al., 2016) and not entirely consistent across the reactions studied. Although the data suggests that the rate coefficient might increase by as much as a factor 10, we do not propose a correction factor until more data becomes available.

### 3.4.3 Migration of hydroxyl-H-atoms

H-migration of a non-enolic hydroxy-H-atom has been considered in many studies (e.g. this work, Asatryan et al., 2010; Fuchs et al., 2018; Jørgensen et al., 2016; Kuwata et al., 2007; Møller et al., 2019; Peeters et al., 2009; Piletic et al., 2019). The rate coefficients tend to be fairly low, $k(298 \text{ K}) \sim 1 \times 10^{-3}$ s$^{-1}$, and the reaction plays only a very small role in the atmosphere even in pristine conditions. Furthermore, the reverse reaction, i.e. the abstraction of a hydroperoxide OOH atom by an alkoxy radical, is rather fast, typically $\geq 10^{10}$ s$^{-1}$, such that even if the reaction occurs, the regeneration of the original peroxy radical is often the main fate. Some exceptions to this reaction reversal have been reported though, such as the Waddington mechanism where the H-shift is followed by fast decomposition of the resulting alkoxy radical (Lizardo-Huerta et al., 2016; Ray et al., 1973a, 1973b). For example, Peeters et al. (Peeters et al., 2009) showed that 1,5-H-migration in hydroxy-isoprene-peroxy radicals could not be reversed due to immediate dissociation of the β-unsaturated-β-OOH-alkoxy radical products. Given the generally low importance of hydroxy-H-migrations, we do not derive a SAR for this reaction class at this time.

The abstraction of an enolic hydrogen atom, C=C–OH, forms a vinoxy-stabilized product radical, C•–C=O, which is more stable than the alkoxy products discussed above, thus preventing rapid reversal of the H-migration. The rate coefficient is significantly faster than the reactions above, $k(298\text{K}) \geq 10^{4}$ s$^{-1}$ for a 1,6-enol H-migration (Peeters and Nguyen, 2012). A recent review by Jenkin et al. (2019) recommends $k(T) = 0.24 \times T^{4.1} \times \exp(-2700\text{K/T})$.

Finally, we should note that α-OH peroxy radicals can undergo an HO$_2$ elimination, forming a carbonyl compound with an HO$_2$ co-product. Jenkin et al. (2019) tabulates these eliminations as a 1,4-H-migration, but mechanistically these reactions occur by a concerted elimination characterized by a weakened, long C–OO bond in the transition state. These reactions have been studied in detail both theoretically and experimentally (Hermans et al., 2004, 2005b, 2005a; Linguerri et al., 2017; Tomas et al., 2001; Veyret et al., 1989). The recent review by Jenkin et al. (2019) has recommendations for tertiary and quaternary carbons based on these studies; for secondary carbons, i.e. HO$_2$ elimination from HOCH$_2$OO•, we calculated a rate coefficient of $1.8 \times 10^{3}$ s$^{-1}$, in agreement with these recommendations (see supporting information).

## 3.5 Hydroperoxide-substituted RO$_2$

### 3.5.1 Migration of α-OOH H-atoms

For many of the α-OOH H-migrations, there are significant differences between the different literature sources (this work; Miyoshi, 2011; Mohamed et al., 2018b, 2018a; Otkjær et al., 2018; Praske et al., 2019; Sharma et al., 2010), of up to several orders of magnitude for some of the reactions considered, though for some of the reactions near-identical values are found





by all authors. This scatter reduces the reliability of the SAR predictions significantly, and in many cases the k(298 K) geometric average of the literature rate coefficients will have an uncertainty exceeding an order of magnitude. Furthermore, most of the data sets are derived at high temperatures, ≥ 300 K, and do not reproduce the curvature in the Arrhenius plot that

is evident in our predictions down to 200 K. To improve the prediction of the rate coefficients below room temperature, we apply the temperature-dependence of our work (α-OOH 1,6-H-migrations) to all 1,5 through 1,7-H-migrations. This does not affect the SAR rate coefficient above 300 K appreciably, and thus remains in agreement with the literature values at higher temperatures. 1,4-H-migrations have high barriers and their Arrhenius plots are more linear due to different tunneling properties; we do not apply the k(T) curvature as above but solely rely on the geometric average of the reported k(T) values.

The product α-OOH alkyl radical formed after migration of the α-OOH hydrogen atom is unstable, and falls apart spontaneously to a carbonyl compound with an OH radical co-product; the addition of $O_2$ on this radical is not competitive (Vereecken et al., 2004).

### 3.5.2 Migration of hydroperoxide H-atoms (scrambling)

The available theoretical data on migration of hydroperoxide-H-atoms can be divided into two groups. On one hand are the

studies by Miyoshi (2011) and Mohamed et al. (2018b), who find comparatively slow H-atom exchange between hydroperoxide group and the –OO$^\bullet$ radical, with rate coefficients of the order of 1 s$^{-1}$ to 10$^{-5}$ s$^{-1}$, depending on the migration span (1,6- to 1,8-H-migration) and substitution around the oxygenated moieties. These calculations are based on single-conformer CBS-QB3 calculations with explicit treatment of internal rotation. Our own calculations (Novelli et al., 2019; Nozière and Vereecken, 2019), however, find rate coefficients ≥ 10$^2$ s$^{-1}$, as do theoretical calculations by Jørgensen et al.

(2016), Møller et al. (2019), and Praske et al. (2019), who find k(298 K) of 10$^2$ to 10$^6$ s$^{-1}$, for a large set of H-exchange reactions. These latter group of calculations are all based on multi-conformer TST calculations based on higher-level CCSD(T)//M06-2X or CCSD(T)-F12//ωB97X-D calculations. The reason for this dichotomy is not evident, and requires further analysis. Experimental data by Praske et al. (2019), constrained by observed products as a function of the competing reaction with NO, finds rate coefficients k(296 K) ~ 10$^2$ s$^{-1}$ for a 1,6-H-migration, and even k(296 K) > 10$^4$ s$^{-1}$ for a 1,7-H-

migration, corroborating the higher values in the second group of theoretical data. Unfortunately, the study by Miyoshi (2011) is the only extensive, systematic study of the scrambling reactions (1,6- through 1,9-H-migration, for all 9 substitution patterns around the active sites). Nozière and Vereecken (2019) pragmatically scaled up the rate coefficients by Miyoshi by a factor 10$^4$, the average difference between the Miyoshi predictions and the observations of Praske et al. (2019). In this work, we apply a similar procedure, where we use the systematic study of Miyoshi (2011) as a basis, but

scale the rate coefficients at 298 K for 1,6-H-migration upwards by a factor 4.8×10$^2$, as obtained by taking the ratio of the experimental value of Praske et al. (2019) of 10$^2$ s$^{-1}$, and the theoretical 2.1×10$^{-1}$ s$^{-1}$ (under)estimate by Miyoshi. Likewise, 1,7-H-migrations k(298 K) are scaled up by a factor 5.6×10$^3$, based on the experimental lower limit of 10$^4$ s$^{-1}$ of Praske et al. and the corresponding value of Miyoshi. Theoretically predicted rate coefficients are typically another factor ~10$^2$ higher, suggesting that we will underestimate H-scrambling rates. Finally, 1,8- and 1,9-H-migrations are scaled up by a factor

1.6×10$^5$, by comparing the predictions from Miyoshi against the Nozière and Vereecken (2019) predictions for 1,8-H-migrations. Additionally, we apply the temperature dependence derived from these latter calculations to incorporate the




impact of entropy and tunneling on the rate coefficient curvature in the low-temperature range. Where available, data by Mohamed et al. (2018b), Praske et al. (2019), and Nozière and Vereecken (2019) are included when calculating the rate coefficient.

For 1,5-H-migration no data is available; this implies gemini OOH/OO$^\bullet$ groups for which there are no obvious atmospheric formation pathway, so we ignore this class of scrambling reactions at this time.

The above scaling procedure is, overall, rather unsatisfactory, and it is clear that more theoretical and experimental data is needed to reduce the uncertainties and avoid the poorly founded corrections. One redeeming factor, however, is that the experimental data and much of the theoretical calculations suggest that H-scrambling reactions across –OOH and –OO$^\bullet$

groups are significantly faster than most, if not all, other RO$_2$ H-migration reactions. Hence, for many practical purposes, the hydroperoxide hydrogens can be considered mobile at the timescale of other product-forming H-migration reactions, and the reactants can be merged into a single reactant pool reacting at bulk reaction rates. This is discussed in more detail in Sect. 5.

**3.6 Carboxylic acid-substituted RO$_2$**

The migration of acidic H-atoms was shown by da Silva (2010b) to be a potential route to OH regeneration in the atmosphere. The migration of the acid H-atom is aided by the formation of a resonance-stabilized acyloxy radical, –C(=O)O$^\bullet$, which readily eliminates CO$_2$, thus preventing reversal of the H-migration. In the case of a β-peroxy radical, e.g. $^\bullet$OOCH$_2$C(=O)OH, the resulting α-OOH alkyl radical product is even intrinsically unstable, spontaneously falling apart to CO$_2$, an OH radical, and a carbonyl compound (da Silva, 2010b; Vereecken et al., 2004). Our own calculations on ethanoic

acid-2-peroxy and 3-hydroxy-2-methyl-propanoic acid-2-peroxy radicals (derived from methacrylic acid by addition of OH and O$_2$) are in good agreement with the data by da Silva (2010b) on ethanoic acid-2-peroxy, propanoic acid-2-peroxy, and 3-hydroxy-propanoic acid-2-peroxy radicals (derived from acrylic acid by addition of OH and O$_2$), with rate coefficients ranging from $3\times10^{-2}$ s$^{-1}$ to 6 s$^{-1}$. As only information is available on β-carboxylperoxy radicals, we can not derive generally applicable SAR expressions at this time; studies on the atmospheric oxidation of (unsaturated) carboxylic acids could

benefit from more information on this pathway.

**3.7 α-nitrate-substituted RO$_2$**

The data set of Otkjær et al. (2018) contains a number of α-ONO$_2$-substituted H-migrations, while our data set (see supporting information) has some information for unsaturated nitrated RO$_2$ derived from isoprene + NO$_3$ + O$_2$ reactions. Compared to an alkyl group, a nitrate group reduces the rate coefficient somewhat, in agreement with e.g. the known

inhibiting effect for H-abstraction by OH radicals from nitrate-substituted VOCs. The inhibition is similar for H-migrations across all spans in both aliphatic and unsaturated RO$_2$, ranging from 0.001 to 0.9 at room temperature, with an average across the data sets of a factor ~0.04. We include this effect in the SAR by a temperature-dependent correction factor $k_{NO3}(T) = k_{aliphatic}(T) \times \exp(-950 \text{ K}/T)$ (see Table 5)



The product α-ONO$_2$ alkyl radical formed after migration of the α-nitrate hydrogen atom is unstable, and falls apart

spontaneously to a carbonyl compound with an NO$_2$ radical co-product; the addition of O$_2$ on this radical is not competitive

(Vereecken, 2008).

**3.8 Other substitutions**

Otkjær et al. (2018) has some information on F-atom substitution for a range of H-migrations; at this time we do not include

halogen atom substitution in our SAR. We have not found systematic data on other relevant substitutions such as the

nitrogen oxides –NO, –NO$_2$, –ONO, –OONO, –OONO$_2$, nor for peroxides –OOR. Otkjær et al. (2018) lists a set of rate

coefficients for ether substitution on the migrating H-atom, α-OR, finding rate coefficients within a factor of 3 up or down

of the corresponding α-OH-substituted H-migration. At this time, we recommend using the k$_{αOH}$(T) values for these α-OR-

substituted H-atoms (see Table 5).

**4 SAR validation and scope of applicability**

Aside from a comparison against the scarce experimental data, it is not overly useful to examine the agreement between the

SAR predictions and the data used to derive the SAR, as the SAR parameters are typically derived using only a few data

points, ensuring good apparent agreement between SAR and training data. Occasionally, the underlying data has been

intentionally strongly corrected (e.g. H-scrambling), such that the predictions will not match the data. Hence, the SAR

parameters are subject more to the (sometimes large) uncertainties on the training set data points than the fitting procedure,

making a visual comparison or statistical goodness-of-fit analysis unrewarding. It is more useful to validate the SAR to its

robustness, examining the reproduction of data points that were not part of the training data; we refer to Vereecken et al.

(2018) for a more in-depth discussion of SAR validation needs in atmospheric chemistry. In this work, we specifically aim

to test the scope of applicability of the SAR, applying it to a test set of over 100 data points that were not in the training set,

and that furthermore are multi-functionalized and thus represent data points that are outside the coverage of the training set.

We distinguish between three groups of data. The first are rate coefficients for reactions that formally work on multi-

functionalized RO$_2$, but where only one functionality is expected to influence the reactivity strongly; the impact of all other

functionalities, or of substituents that are not included in the SAR, are neglected ("spectator" groups). A second group of

data are those where multiple functionalities affect the reaction rate, e.g. two substituents located in α- and β-position of the

migrating H-atom. Thirdly, we visualize the results for cyclic species, where the H-migration involves a peroxy radical or

migrating H-atom on the parent molecule cycle, or where the migration spans carbons part of the (mono- or multi-cyclic)

ring structure. The resulting changes in ring strain or geometric constraints are not accounted for in the SAR, and we

explicitly expect the SAR to perform poorly for such compounds.

**4.1 Comparison against experimental data**

Table 9 lists the available experimental rate coefficients, and compares them to the SAR predictions at the appropriate

temperature, accounting for multiple channels where necessary. For 21 of the 22 available values, the performance is





excellent, reproducing the values on average by a factor of 2 (see Figure 3). Only a single. multi-functionalized molecule, •OO-CH(CH₃)-C(=O)-CH(OOH)-CH₃, does not match the predictions, by a large factor ~$10^2$. The direct theoretical calculations of 0.25 and 0.54 s⁻¹ by Møller et al. (2019) conform to the lower limit found by Crounse et al. (2013), ≥ 0.1 s⁻¹, so clearly the SAR does not behave as desired for this compound. As our SAR matches the theoretical predictions for α-

OOH-substituted RO₂ well, the likely culprit for the failure is the SAR correction for endo-β-oxo 1,5-H-migrations (see Table 5). It is the only β-oxo correction to slow down the reaction, and was derived from a single data point, obtained by a rel-MC-TST calculations using only 3 conformers, making it one of the least reliable values used the SAR. An alternative source of error could be that the SAR should account for interactions between β-oxo and α-OOH substitutions. Additional training data is needed to resolve this issue.


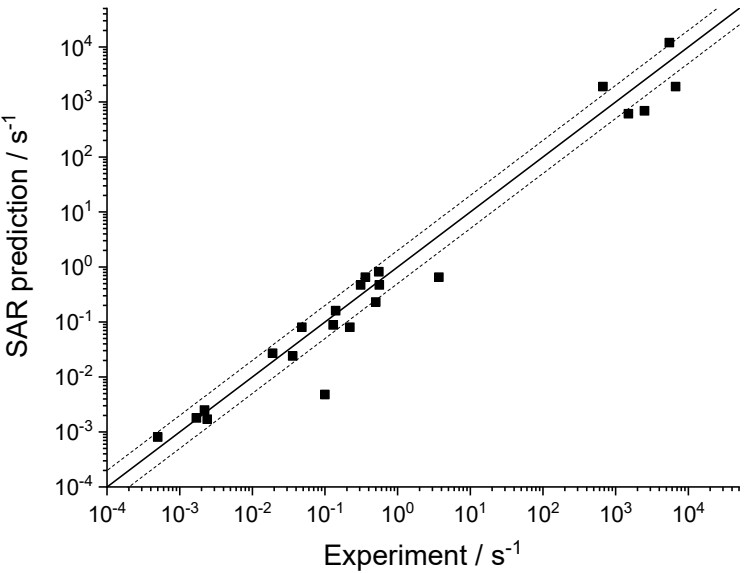

*Figure 3: Comparison of the experimental values against the SAR predictions (various temperatures). The solid line shows a 1:1 agreement, with the dotted lines delineating a factor 2 uncertainty.*



### 3.2 Performance with spectator substitution

To keep the number of SAR parameters manageable, and to make the SAR easily applicable, one hopes that each SAR class
is applicable to a wide range of compounds, without having to resort to explicitly examining all possible permutations of
functionalities on the reactant. Due to the nature of the title reaction, a fairly large number of SAR classes is unavoidable, as
the reaction acts from short to long range, and is dependent on two reactive sites (–OO$^\bullet$ moiety and migrating H-atom) that
can be influenced by their neighbors (even long-range neighbors in case of delocalization). In this test set, we include those
compounds that have non-aliphatic "spectator" substituents located such that they are not expected to have an overly large
effect on the H-migration reaction. Examples include substituents outside the TS cycle, or located sufficiently far from both
reaction sites.

Figure 4 shows the agreement between the SAR predictions and our theory-based data set (see supporting information), for
those reactions of non-cyclic RO$_2$ not included in the training set. The performance is comparatively good, with most
predictions within an order of magnitude. Given that this set includes stereo-isomers where direct calculations show rate
coefficient differences of almost 2 orders of magnitude, this indicates that the SAR performs well, and that spectator
functionalities can be well approximated by aliphatic substituents. The largest scatter is found for a very slow H-migration,
$k_{theory}$(298K)~ $10^{-10}$ s$^{-1}$, which is not relevant in the atmosphere, and for H-scrambling reactions, which have been discussed
earlier to suffer from large discrepancies between the predictions in the various literature sources. As expected (see above),
we typically under-predict H-scrambling rates; despite this we still find that the scrambling reactions are several orders of
magnitude faster than other H-migrations, supporting pooling hydroperoxide-substituted RO$_2$ radicals, as discussed in
section 5. A similar analysis of selected literature data is shown in Figure 5 (D'Ambro et al., 2017; Jørgensen et al., 2016;
Knap et al., 2015, 2016; Knap and Jørgensen, 2017; Kurtén et al., 2015; Møller et al., 2016, 2019; Piletic et al., 2019;
Praske et al., 2018, 2019; Wang et al., 2018; Xing et al., 2018). The SAR predictions show a significantly wider scatter than
for the previous test set. However, one should take into account that this data is based on a wider range of theoretical
methodologies, and the wider scatter is at least partly caused by the use of MC-TST calculations based on a limited number
of conformers, or from the use of DFT-based energies which carry a larger uncertainty. At this point, we anticipate that the
scatter relative to the SAR prediction will decrease strongly when applying better quantum chemical methodologies, and by
incorporating more conformers in the kinetic analysis, either by full MC-TST or rel-MC-TST.



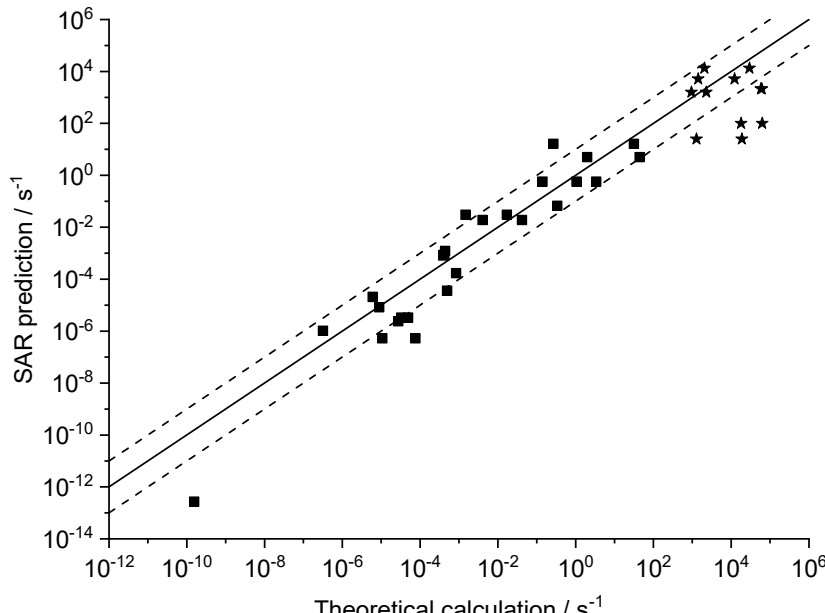

*Figure 4: k(298 K) H-migration in multi-functionalized species with spectator substituents. The data shown are derived in this work or our earlier publications, using MC-TST incorporating all conformers compared against the SAR prediction. The stars indicate OOH/OO• H-scrambling reactions. The solid line shows 1:1 correspondence, with the dotted lines delineating an order of magnitude uncertainty.*


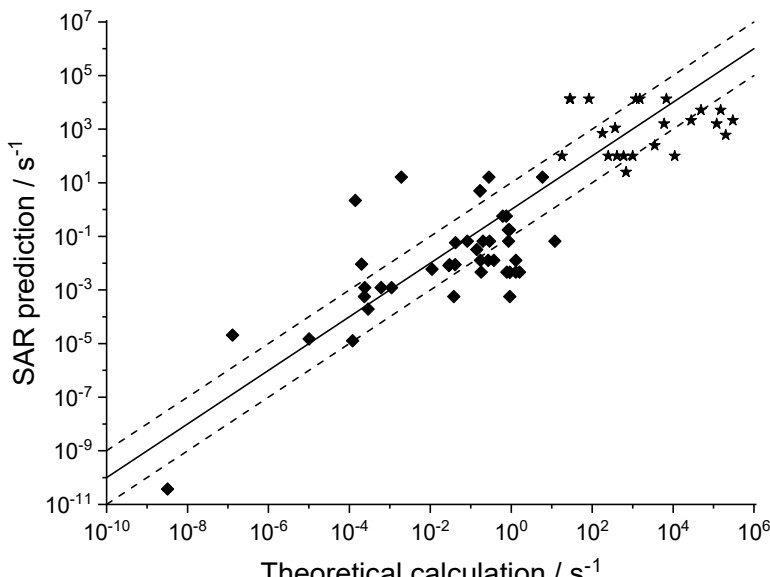

*Figure 5: Literature theory-based k(298 K) rate coefficients compared against the SAR predictions (see text for references). The stars are OOH/ OO• H-scrambling reactions. The solid line shows 1:1 correspondence, with the dotted lines delineating an order of magnitude uncertainty.*


### 4.3 Multi-functionalized compounds

Our SAR currently does not directly support the prediction of rate coefficients for $RO_2$ radicals when the H-migration is affected by more than one functionality, due to the lack of data to derive reliable cross terms accommodating two or more simultaneous functionalities. As a test case, however, we examine here the 1,6-H-migration in methyl-4-hydroxy-2-butene-

1-peroxy, i.e. the Z-δ-1-OH-isoprene-4-peroxy (Z-HO-CH₂-C(CH₃)=CH-CH₂OO•, Z-δ-RO₂-1) and Z-δ-4-OH-isoprene-1-peroxy (Z-•OO-CH₂-C(CH₃)=CH-CH₂OH, Z-δ-RO₂-4) radicals of critical importance in the isoprene oxidation (Bianchi et al., 2019; Nguyen et al., 2010; Novelli et al., 2019; Peeters et al., 2009, 2014; Wennberg et al., 2018). Teng et al. (2017) derived rate coefficients for these unimolecular reactions from experiments, finding rates of $0.36 \pm 0.14$ s⁻¹ (Z-δ-RO₂-1) and $3.7 \pm 1.0$ s⁻¹ (Z-δ-RO₂-4), in good agreement with the theoretical data, e.g. 0.49 s⁻¹ and 5.4 s⁻¹ as derived by Peeters et al.

(2014) for the LIM1 mechanism. These values have been incorporated in atmospheric models (e.g. Wennberg et al., 2018), though some models use lower rate coefficients; the MCM v3.3.1 (Jenkin et al., 2015) for example uses 0.11 and 1.2 s⁻¹, respectively. The overall kinetics is mostly sensitive (Novelli et al., 2019) to the ratio of the H-migration rate over $O_2$-





elimination and other reactions, whose rate coefficients also different in these models. The rate coefficient for 1,6-H-migration is significantly faster than in the corresponding aliphatic compounds, and is accelerated by the formation of a stable α-OH allyl radical. The main impact is due to the allylic resonance, which enhances the rate by over a factor 700 (see Table 2); the α-OH substitution has an impact of a factor of ~10 in our SAR (see Table 6). A SAR prediction including only the TS endo-cyclic double bond, 0.65 s$^{-1}$, reproduces the experimental and direct theoretical results within a factor of 6 to 8; bearing in mind that our SAR does not distinguish between substitutions on the second allylic radical site, which in this case has an impact of a factor 10, this is remarkably good. Applying corrections for both the allylic and α-OH functionalities, yielding 6.5 s$^{-1}$, apparently overestimates the rate coefficients by up to a factor 20. This test suggests that there is mutual influence between multiple functionalities beyond a purely additive or multiplicative combination. However, within the uncertainties of our SAR, i.e. allowing for the scatter on the underlying data and the simplifications in the SAR such as ignoring stereochemistry and secondary allylic sites, we would be hard-pressed to make a case for including a second-order bi-functional correction on the basic SAR at this time.

Figure 6 shows an analysis of available rate coefficients on multi-functionalized species (This work ; Knap and Jørgensen, 2017; Møller et al., 2019; Peeters et al., 2009), approximating the rate coefficient using only the existing SAR parameters for mono-functional RO$_2$ radicals. We find that the agreement is generally good; the large scatter on the Knap and Jørgensen (2017) data appears due to the use of lower-level, DFT-based reaction energies.

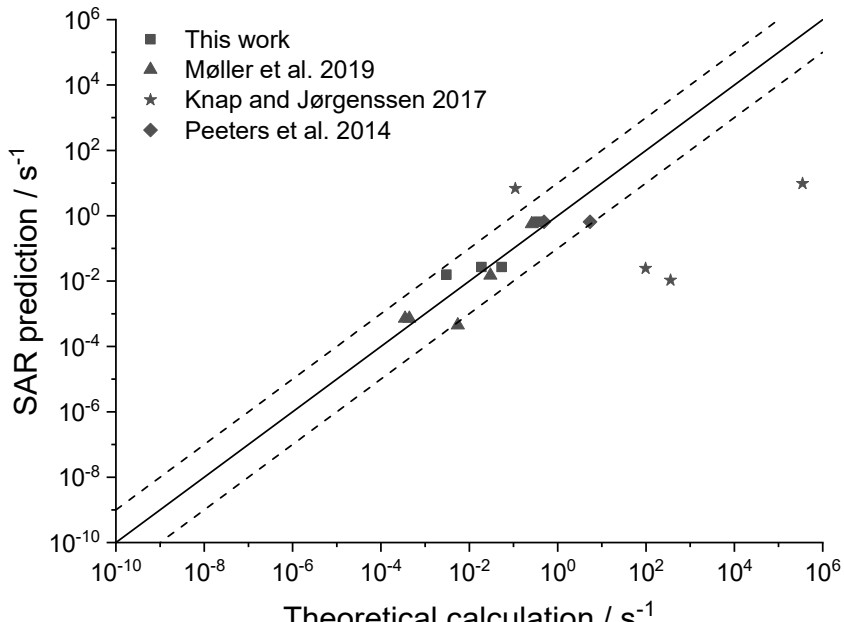

*Figure 6: Theory-based k(298 K) rate coefficients for multi-functionalized RO₂ compared against the SAR predictions. The solid line shows 1:1 correspondence, with the dotted lines delineating an order of magnitude uncertainty.*

### 4.4 Cyclic structures

For completeness, Figure 7 depicts the SAR performance for a set of H-migrations in cyclic species, such as substituted menthenyl-peroxy radicals, pinonaldehyde-peroxy radicals, and bicyclic peroxide-peroxy radicals (This work ; Kurtén et al., 2015; Xu et al., 2019). As the SAR has no provisions for strained bicyclic structures, these predictions are not expected to be particularly good.

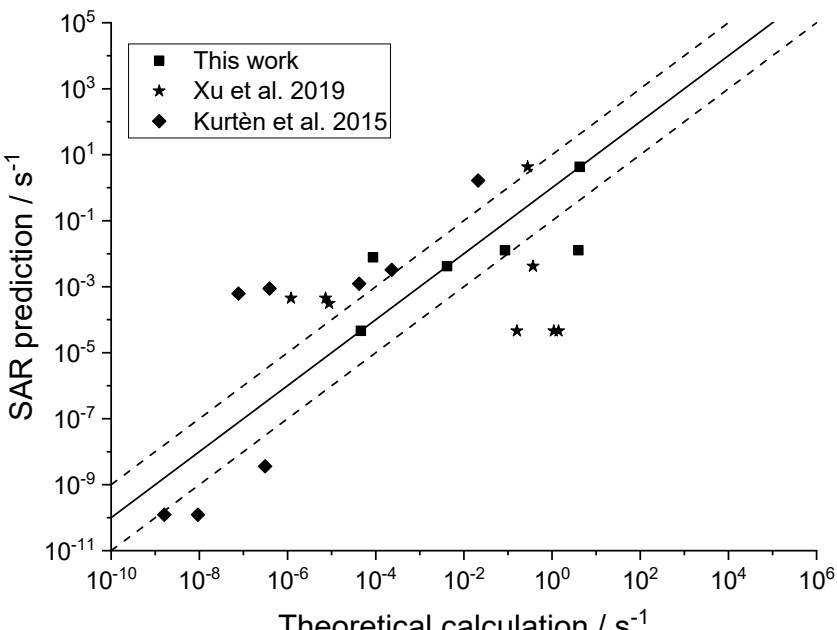

*Figure 7: Theory-based k(298 K) rate coefficients for cyclic RO₂ compared against the SAR predictions. The solid line shows 1:1 correspondence, with the dotted lines delineating an order of magnitude uncertainty.*

**4.5 Untestable SAR parameters**

Much of the uncertainty on the SAR predictions are not shown in the above tests. In particular, the values in the SAR that are inferred only by relative rate considerations —either the temperature-dependence $k(T)$ and/or absolute value at 298 K— can not be tested due to lack of data for comparison. While the other SAR predictions are seen above to be somehow "average" to the available data, albeit possibly with a large uncertainty, a critical difference with the guestimated values is that these could have a (large) systematic bias that influences how we perceive the importance of these reactions. In this respect, it is important that further systematic studies are performed to provide a reference value in each of the SAR categories. Until then, we rely solely on the assumption that the reactivity trends are applicable across SAR classes.

**5. Implications of rapid H-atom scrambling in HOO-RO₂**

The rapid migration of hydroperoxide-H-atoms in HOO-substituted peroxy radicals has some interesting ramifications for the chemistry and kinetics of these radicals. As already earlier, the scrambling reactions are typically several orders of magnitude faster than other RO₂ H-migration reactions. As a result, the HOO-RO₂ isomeric population wil readily





equilibrate, and re-equilibrate to compensate for loss (or gain) of specific isomers; for the reaction rates at play, this
typically means that the isomers are present in their Boltzmann equilibrium relative concentrations. It then becomes
convenient to describe the isomers as a single equilibrated pool of HOO-RO$_2$ molecules (see Figure 8). As the effective
contribution of an isomer is then no longer equal to the total concentration (or its original concentration implied by any
formation processes), the elementary rate coefficient is not a good measure of the rate of product formation unless the
isomer-specific contribution is explicitly accounted for. This is done by "bulk" rate coefficients, operating on the pool of
HOO-RO$_2$ molecules.

For a specific elementary reaction, one then finds a bulk rate coefficient as:

$$k_{bulk}(T) = \frac{k_1(T) Q_1(T) \exp\left(\dfrac{-E_1}{kT}\right)}{Q_1(T) \exp\left(\dfrac{-E_1}{kT}\right) + Q_2(T) \exp\left(\dfrac{-E_2}{kT}\right) + \ldots} \tag{E1}$$

which can be used to convert elementary rate coefficients to population-weighted rate coefficients (Vereecken et al., 2017),
with $Q_i(T)$ the partition functions of each of the isomers, $E_i$ their relative energy, and $k_1$ the elementary rate coefficient for a
reaction starting from isomer 1. If a product is formed in more than one channel, the bulk rate coefficients are summed.
Using the equilibrium constant expressed as a ratio of partition functions and relative energies, and using the resulting
contributing fractions of each isomer in the total concentration, we can rewrite this equation as follows:

$$k_{bulk}(T) = f_i(T) \cdot k_1(T) \tag{E2}$$

where $f_i(T)$ is the temperature-dependent fraction of reactant $i$ in the isomer pool and undergoing reaction 1 with elementary
rate coefficient $k_1(T)$.

These formulae imply that the effective, bulk rate coefficient is less than the elementary rate coefficients obtained by the
theoretical rate calculations and SAR predictions for a single isomer. Bulk rate coefficients have been described and applied
earlier, both based on HOO-RO$_2$ scrambling and other RO$_2$-equilibrating mechanisms (Novelli et al., 2019; Nozière and
Vereecken, 2019; Peeters et al., 2009, 2014), and have proven necessary to connect the elementary reaction kinetics to the
phenomenological observations.


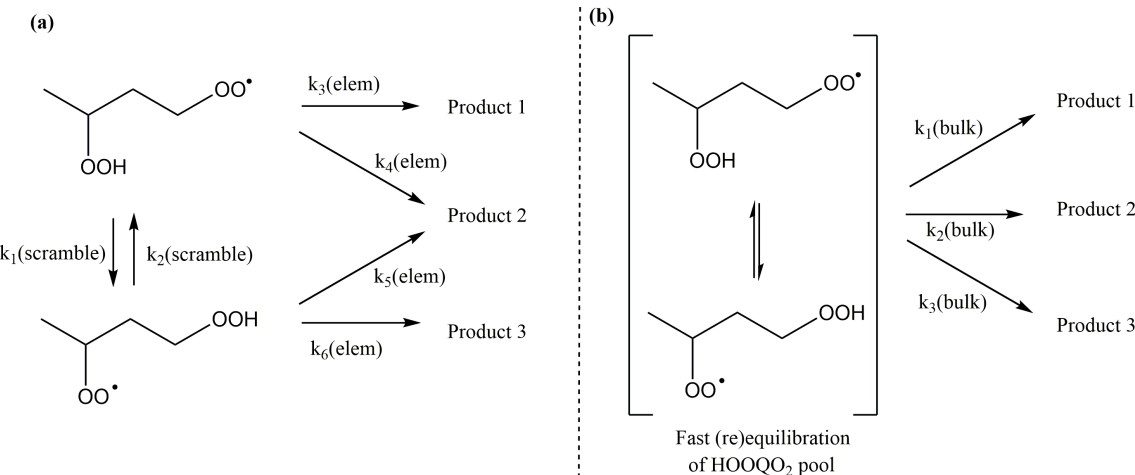

*Figure 8: Illustration of fast scrambling and the use of bulk rate coefficients to reduce the complexity of the kinetic model, with (a) the explicit model, and (b) the model using a single lumped pool (in square brackets). In the example above, $k_2$(bulk) is the sum of the bulk rate coefficients for each of the two channels. The scheme is trivially extended to more isomers and reaction channels.*

The ability to describe the pool of HOO-RO$_2$ as a single species in a kinetic model could lead to significant simplification, and a more natural implementation for comparison against experiment. For studying the formation of highly oxidized molecules (HOM), typically multi-OOH-substituted molecules formed through multiple RO$_2$-H-migrations + O$_2$ addition sequences, the pooling could provide a more direct link to aerosol models, which often lump species in similarity classes. Unfortunately, there is no simple method to predict the relative contributions of the HOO-RO$_2$ in a pool and thus obtain

$k_{pool}$(T). A pragmatic solution proposed by Nozière and Vereecken (2019) involves simply counting the number of isomers *n* contributing to the pool, assigning an equal population fraction $f_i = 1/n$ to each, and using this as a measure for the bulk rate coefficient ratio relative to the elementary rate coefficient $k_j$(T) :

$$k_{pool}(T) = \frac{1 \cdot k_{elem}(T)}{\sum_i 1} = \frac{k_{elem}(T)}{n} \qquad (E3)$$

This solution is readily implemented both manually and in automated mechanisms, but it implies that the isomers should be

present in roughly equal concentrations. In some cases, one/some of the isomers could be dominant or negligible, e.g. due to stabilization by H-bonding, or by formation of more stable hydroperoxides (e.g. peracids). The simplified approach above thus carries a larger uncertainty if the isomers are not sufficiently "similar".

A corollary of the fast re-equilibration is that the HOO-RO$_2$ molecules have access to all exit channels in the pool, irrespective of the isomer that was originally formed. An example of such a system has been recently described by Novelli





et al. (2019), where the initially formed isomer of di-HPCARP-RO$_2$, a dihydroperoxy-aldehyde-peroxy radical formed from isoprene (3 isomers per pool), does not have access to a fast aldehyde-H-migration pathway (only a 1,4-H-shift), but fast scrambling allowed access to much faster 1,5- and 1,6-aldehyde-H-migration channels, leading to acyl radical formation as the dominant channel. Again, this could allow reductions in mechanism complexity, as less favorable exit channels for some of the isomers can be omitted in favor of dominant exit channels through other isomers in the HOO-RO$_2$ pool. Such

mechanism reductions are readily implemented.

Unfortunately, there are conditions where the reduction of the mechanism by pooling isomers is unlikely to work. An example is a mechanism that should be applicable across a large range of RO$_2$ co-reactant (e.g. NO) concentrations. At low concentrations where the bimolecular reaction is slower than the H-atom scrambling, use of an HOO-RO$_2$ pool will be appropriate, but for increasingly high concentrations the bimolecular reaction will compete with and eventually overwhelm

the equilibration by H-scrambling, leading to product yields that are determined by the nascent isomeric contributions rather than the equilibrium contributions. For such broadly applicable models, an explicit multi-conformer mechanism must be constructed (as in panel (a) of Figure 8), though some model reduction may still be possible by omission of some of the slower H-migration channels.

**6. Experimental verification of peroxy radical reaction mechanisms**

In addition to RO$_2$ and HOOQO$_2$, a number of stable products were detected in the experiments described earlier by Nozière and Vereecken (2019). For all the RO$_2$ studied, the carbonyl compound corresponding to initially formed RO$_2$ was observed and represented by far the main reaction product, evidencing the importance of the keto-hydroxy channel in the self-reaction of RO$_2$ in all cases. These products were mostly observed at mass M+19, and at lower intensities at M+37 and sometimes even M+55. Thus, in the 1-butylO$_2$ system, butanal (MW = 72 g mol$^{-1}$) was mostly observed at m/z = 91 and 109, in the 1-

pentylO$_2$ system, pentanal (MW = 86 g mol$^{-1}$) at m/z = 105 and 123, in the 1-hexylO$_2$ system, hexanal (MW = 100 g mol$^{-1}$) at m/z = 119 and 137, in the ethylhexylO$_2$ system, ethylhexanal (MW = 128 g mol$^{-1}$) was observed at m/z = 147. For dimethylhexylO$_2$, 2,5-dimethyl-hexanal and isomers (MW = 128 g mol$^{-1}$) were mostly observed at M+1 (m/z = 129) and M+19 (m/z = 147). Note, however, that the technique does not distinguish between isomers, therefore the position of the carbonyl groups in these compounds (and of the other substituents in the other types of products discussed below) could not

be confirmed. The ROOH hydroperoxides produced by the RO$_2$ + HO$_2$ reactions were also observed in most systems, but their main peak at M+19 (Nozière and Hanson, 2017) overlapped with the M+37 peaks for the main carbonyl products: butylhydroperoxide, MW = 90 g mol$^{-1}$ and m/z$_{M+19}$ = 109; pentylhydroperoxide, MW = 104 g mol$^{-1}$ and m/z$_{M+19}$ = 123; hexylhydroperoxide, MW = 118 g mol$^{-1}$ and m/z$_{M+19}$ = 137; 2,5-dimethyl-hexylhydroperoxide, MW = 146 g mol$^{-1}$ and m/z$_{M+19}$ = 165. The contribution of the hydroperoxides to these signals was thus determined by adding NO periodically in

the reactor. The systematic decrease of these signals in the presence of NO confirmed the significant contribution of hydroperoxides, as these compounds are formed only in the absence of NO. In the 1-butylO$_2$, 1-pentylO$_2$ and 1-hexylO$_2$ systems, peaks corresponding the alcohols produced by the self-reaction of RO$_2$ were also observed at their M+19 and M+37 peaks: in the 1-butylO$_2$ system butanol (MW = 74 g mol$^{-1}$) was thus observed at m/z = 93 and 111, in the 1-pentylO$_2$





system, pentanol (MW = 88 g mol$^{-1}$) at m/z = 107 and 125, and in the 1-hexylO$_2$ system, hexanol (MW = 102 g mol$^{-1}$) was

observed at m/z = 121 and 139. These peaks were found to disappear systematically when NO was added to the reactor, confirming that they were produced by the reaction of the RO$_2$ radicals in the absence of NO, in agreement to known RO$_2$ + RO$_2$ chemistry. Finally, in the 1-pentylO$_2$, 1-hexylO$_2$ and dimethylhexylO$_2$ systems, peaks corresponding to the carbonyl analogs of HOOQO2 were observed. Interestingly, where the HOOQO$_2$ radicals were exclusively observed at their M+55 peaks due to water clustering (Nozière and Vereecken, 2019), these carbonyls were also observed at their M+55 and even

M+73 peaks. In the 1-pentylO$_2$ system, pentanone hydroperoxide (MW = 118 g mol$^{-1}$) was thus observed at m/z = 173 and 191; in the 1-hexylO$_2$ system, hexanone hydroperoxide (MW = 132 g mol$^{-1}$) was observed at m/z = 187 and 205, and in the dimethylhexylO$_2$ system dimethylhexanone hydroperoxide (MW = 146 g mol$^{-1}$) was observed at m/z = 201 and 219. The formation of these carbonyl compounds was consistent with the fast decomposition of the α-OOH alkyl radicals produced by the fast H-migration from the HOOQO$_2$ (pathway R4 shown in introduction). The observation of all these products is

illustrated for the case of the 1-pentylO$_2$ system in Figure 9 and Figure 10. These products thus confirmed the main expected reaction pathways for RO$_2$ and HOOQO$_2$.

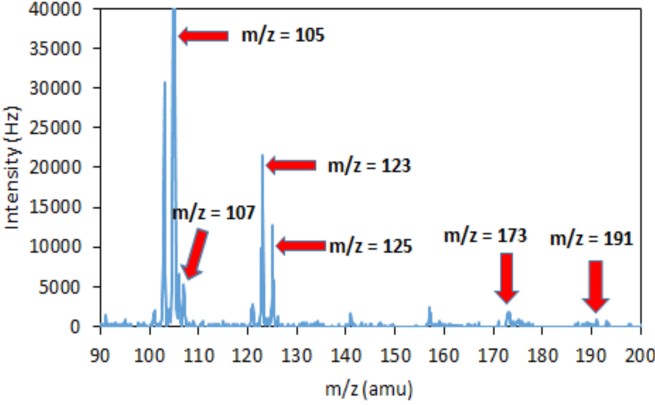

*Figure 9: Main product peaks in the 1-pentylO2 system (experiments of 08/11/18). m/z 105 is C$_5$H$_{10}$=O ; m/z 107 is C$_5$H$_{11}$OH; m/z 123 is mostly C$_5$H$_{11}$OOH with some contribution of C$_5$H$_{10}$=O; m/z 125 is C$_5$H$_{11}$OH ; m/z 173 and m/z 191 are HOOC$_5$H$_9$=O.*






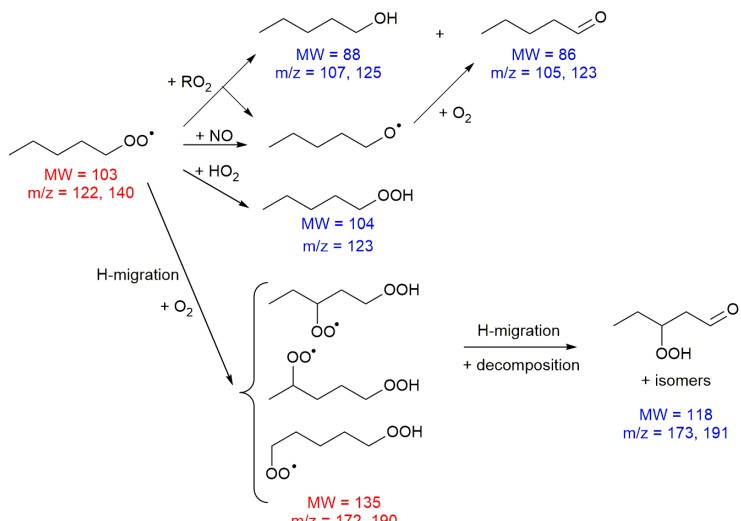

*Figure 10: Reaction mechanisms for the 1-pentylO2 system showing the radicals (MW and m/z in red) and stable products (MW and m/z in blue) observed experimentally.*

**7. Discussion and conclusions**

In this work, an extensive SAR is presented for predicting the temperature-dependent rate coefficient for H-migration in substituted alkylperoxy radicals. The SAR is mostly based on theoretical data at various levels of theory, using both newly presented and literature data. The theoretical methodology used to predict the rate coefficient is evaluated, taking the best available data into account when deriving the absolute rate coefficient and its temperature dependence, thus leading to the best available estimates. Modified Arrhenius parameters are presented for the temperature range 200-450 K, sufficient to

cover all relevant atmospheric conditions. Experimental data is scarce but is used when available. The SAR generalized earlier work on $RO_2$ H-migration, and can provide valuable guidance for mechanism development in atmospheric conditions. The reliability of the SAR depend strongly on the SAR reaction class. For aliphatic $RO_2$, there is ample theoretical data, which tends to agree well, allowing for robust predictions across the entire temperature range. The experimental data by Nozière and Vereecken on aliphatic $RO_2$ H-migration was not used in the derivation, but provides very

valuable verification of the *a priori* predictions, as aliphatic $RO_2$ serve as the core of the SAR, representing the main features of the carbon backbone of the molecules. Migration spans from 1,4- to 1,8 are considered.

For unsaturated $RO_2$, we have generated a large number of characterizations, and find that allylic stabilization of the alkyl radical greatly enhanced the H-migration rate. The SAR distinguishes between endo- and exo-cyclic double bonds in the


cyclic transition state. Likewise, β-oxo substitution on the migrating H-atom, either endo- or exo-cyclic in the TS, can enhance the rate of migration. Hydroxy and hydroperoxide-substituted $RO_2$ radicals show enhanced abstraction rates for the α-H-atoms. Migration of aldehydic and acidic H-atoms are likewise fast, whereas nitrate-substitution slows down migration of the α-H-atoms. For none of the above SAR classes full coverage is available, with several reaction types that have no (reliable) rate predictions available in the literature. The current SAR derivation thus also serves as a good guide for future studies, indicating where efforts are best applied. A case in point would be the H-atom scrambling in hydroperoxide-

substituted $RO_2$ radicals, for which the available data splits into two groups, predicting rate coefficients separated by 4 orders of magnitude or more. Targeted calculations could prove very valuable to resolve this issue. Other important reactions that could benefit specifically from additional data are the migration of aldehydic H-atoms, and the migration of β-oxo H-atoms, for which very few data is available for some of the migration spans yet have fast reaction rates that could be of importance in the atmosphere. In this work we introduce relative multi-conformers transition state theory, rel-MC-TST, a

technique for reducing the computational cost of systematic rate studies that appears well suited to aid in the characterizing the many reaction rates needed to enhance the current SAR.

Compared to the scarce available experimental data, SAR performs very well, reproducing the data on average within a factor 2, comparable to the experimental uncertainty. While in some cases the experimental data was used in the SAR training data, most of the predictions are based primarily on theoretical data, suggesting that the theoretical data provides a

good basis for the SAR. A further analysis of the performance of the SAR focused on its application to multi-functionalized $RO_2$ radicals, probing the scope of applicability beyond the mono-functionalized training set. By and large, the SAR performs well, showing a scatter relative to the available theoretical data of about an order of magnitude; it should be noted here that the underlying data is itself expected to show scatter of this magnitude, and the SAR can thus not be tested beyond this accuracy. The SAR also suffers somewhat from pragmatic simplifications, such as the lumping of stereo-isomers which

can show rate coefficients differing by up to two orders of magnitude.

In this work, we also report experimental verification of the reaction mechanisms by direct measurements of the products as a function of the reaction conditions; earlier experimental work is often indirect, relying to some extend on mechanistic assumptions to derive the rate coefficients of the elementary or bulk rate coefficients. We discuss in some detail the impact of fast isomerisation reactions by H-scrambling in hydroperoxide-$RO_2$, which affects the apparent rate coefficient and

product distribution. Under conditions with sufficiently slow competing bimolecular reactions such as with NO, $HO_2$ and $RO_2$, the unimolecular decay of such HOO-$RO_2$ is better described using bulk rate coefficients acting on an equilibrated pool of peroxy radicals.

Though we anticipate that the SAR parameters will be improved rapidly over the next few years, the current SAR can already provide guidance in the generation of kinetic models, experimental design, and theoretical studies. Further literature

review can also provide additional points of reference, though the status appears that the published data has quite some scatter for rather similar reactions, and adding more sources may thus not necessarily increase the reliability. Extension and refinement of the SAR should thus be based on combined literature data, new theoretical calculations at high levels of theory, and especially experimental work. We feel improvement for aldehydic H-migration and unsaturated compounds are



most needed, as these reactions have the highest rate coefficients and the functionalities are the most common, affecting

atmospheric chemistry most.

### Supplement

The supplement related to this article is available online, and contains additional information on methodologies, the theoretically derived rate coefficients for ~150 compounds, the description of the rel-MC-TST methodology, a documented spreadsheet implementing the SAR derivation, and the raw quantum chemical data.

**Author contributions**

The quantum chemical calculations, theoretical kinetic calculations, literature study, and SAR analysis were performed by LV. The experiments and their analysis were done by BN. Both authors contributed to the discussion and writing of the manuscript.

### Competing interests

The authors declare that they have no conflict of interest.

### Acknowledgments

LV gratefully acknowledges the Max Planck Graduate Center–Johannes Gutenberg University Mainz (MPGC). He also thanks A. Kiendler-Scharr and A. Wahner (Forschungszentrum Jülich) for supporting this project.

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



*Table 1: Rate coefficients for H-migration in aliphatic peroxy radicals, accounting for all abstractable H-atoms in the group.*

| Substitution pattern | | k(298K) / $s^{-1}$ | k(200-450K) = A × $(T/K)^n$ × exp(-Ea / T) | | |
|---|---|---|---|---|---|
| H-atom | –OO• | | A / $s^{-1}$ | n | $E_a$ / K |
| **1,4-H-migration** | | | | | |
| –$CH_3$ | –$CH_2OO$• | $1.98×10^{-11}$ | 1.21E+07 | 1.4 | 14586 |
| | >CHOO• | $2.13×10^{-13}$ | 2.74E+10 | 0.55 | 16790 |
| | >C(OO•)– | $2.68×10^{-13}$ | 3.01E+10 | 0.6 | 16835 |
| –$CH_2$– | –$CH_2OO$• | $5.86×10^{-9}$ | 1.43E+10 | 0.55 | 13551 |
| | >CHOO• | $3.72×10^{-11}$ | 2.70E+10 | 0.45 | 15078 |
| | >C(OO•)– | $1.23×10^{-10}$ | 6.85E+10 | 0.35 | 14829 |
| –CH< | –$CH_2OO$• | $1.62×10^{-7}$ | 4.01E+10 | 0.4 | 12614 |
| | >CHOO• | $3.661×10^{-9}$ | 6.35E+10 | 0.3 | 13715 |
| | >C(OO•)– | $4.17×10^{-8}$ | 1.84E+11 | 0.2 | 13133 |
| **1,5-H-migration** | | | | | |
| –$CH_3$ | –$CH_2OO$• | $8.22×10^{-6}$ | 1.01E-32 | 13.94 | 5195 |
| | >CHOO• | $1.50×10^{-5}$ | 1.31E-27 | 12.46 | 6010 |
| | >C(OO•)– [a] | $2.22×10^{-4}$ | 7.45E-23 | 10.79 | 5650 |
| –$CH_2$– | –$CH_2OO$• | $8.10×10^{-4}$ | 5.00E-20 | 9.81 | 5534 |
| | >CHOO• | $1.23×10^{-3}$ | 2.95E-24 | 11.26 | 4966 |
| | >C(OO•)– [a] | $4.49×10^{-4}$ | 7.45E-23 | 10.79 | 5440 |
| –CH< | –$CH_2OO$• | $3.50×10^{-2}$ | 1.77E-03 | 4.39 | 6563 |
| | >CHOO• | $5.80×10^{-2}$ | 1.64E-35 | 14.99 | 2420 |
| | >C(OO•)– | $6.20×10^{-4}$ | 2.24E-16 | 8.73 | 6280 |
| **1,6-H-migration** | | | | | |
| –$CH_3$ | –$CH_2OO$• | $4.06×10^{-6}$ | 5.66E-17 | 8.59 | 7137 |
| | >CHOO• | $1.28×10^{-5}$ | 2.22E-22 | 10.52 | 6353 |
| | >C(OO•)– [a] | $2.48×10^{-6}$ | 3.47E-18 | 9.05 | 7235 |
| –$CH_2$– | –$CH_2OO$• | $8.83×10^{-4}$ | 1.68E-12 | 7.19 | 6223 |
| | >CHOO• [a] | $7.83×10^{-4}$ | 3.47E-18 | 9.05 | 5519 |
| | >C(OO•)– [a] | $3.12×10^{-4}$ | 3.47E-18 | 9.05 | 5793 |
| –CH< | –$CH_2OO$• | $4.30×10^{-2}$ | 1.88E-03 | 4.25 | 6285 |
| | >CHOO• | $6.82×10^{-2}$ | 1.28E-35 | 14.71 | 1831 |
| | >C(OO•)– [a] | $2.45×10^{-2}$ | 3.47E-18 | 9.05 | 4493 |


| | | 1,7-H-migration | | | |
|---|---|---|---|---|---|
| $-CH_3$ | $-CH_2OO^\bullet$ | $1.03\times10^{-6}$ | 1.47E-16 | 8.21 | 7186 |
| | $>CHOO^\bullet$ | $1.82\times10^{-6}$ | 4.57E-25 | 10.98 | 5873 |
| | $>C(OO^\bullet)-$ [a] | $1.54\times10^{-8}$ | 2.52E-16 | 8.19 | 8561 |
| $-CH_2-$ | $-CH_2OO^\bullet$ | $9.08\times10^{-5}$ | 3.72E-05 | 4.59 | 7535 |
| | $>CHOO^\bullet$ [a] | $1.93\times10^{-5}$ | 2.52E-16 | 8.19 | 6435 |
| | $>C(OO^\bullet)-$ [a] | $4.32\times10^{-6}$ | 2.52E-16 | 8.19 | 6882 |
| $-CH<$ | $-CH_2OO^\bullet$ | $1.86\times10^{-2}$ | 1.61E-18 | 8.98 | 4218 |
| | $>CHOO^\bullet$ [a] | $1.72\times10^{-3}$ | 2.52E-16 | 8.19 | 5097 |
| | $>C(OO^\bullet)-$ [a] | $3.26\times10^{-4}$ | 2.52E-16 | 8.19 | 5593 |
| | | 1,8-H-migration | | | |
| $-CH_3$ | $-CH_2OO^\bullet$ | $8.60\times10^{-8}$ | 3.89E-27 | 11.4 | 6083 |
| | $>CHOO^\bullet$ [a] | $3.6\times10^{-10}$ | 7.49E-17 | 8.18 | 9307 |
| $-CH_2-$ | $-CH_2OO^\bullet$ | $1.03\times10^{-6}$ | 6.83E-01 | 3.04 | 9150 |
| | $>CHOO^\bullet$ [a] | $6.04\times10^{-8}$ | 7.49E-17 | 8.18 | 7769 |
| $-CH<$ | $-CH_2OO^\bullet$ | $2.44\times10^{-8}$ | 1.58E-22 | 10.09 | 7395 |

[a] Averaged temperature-dependence from other reactions used





*Table 2: Rate coefficients for allylic H-migration in peroxy radicals with an endo double bond within the TS ring, accounting for all abstractable H-atoms in the group.*

| Substitution pattern | | $k(298K)$ / $s^{-1}$ | $k(200\text{-}450K) = A \times (T/K)^n \times \exp(-E_a / T)$ | | |
|---|---|---|---|---|---|
| H-atom | –OO$^\bullet$ | | $A$ / $s^{-1}$ | $n$ | $E_a$ / K |
| **1,5-H-migration** | | | | | |
| =C–CH$_3$ | =CHOO$^\bullet$ | $1.70\times10^{-4}$ | 1.91E-95 | 35.01 | -2963 |
| | =C(OO$^\bullet$)– | $3.04\times10^{-3}$ | 1.02E-89 | 33.22 | -2935 |
| =C–CH$_2$– | =CHOO$^\bullet$ | $1.86\times10^{-2}$ | 2.31E-94 | 34.58 | -4361 |
| | =C(OO$^\bullet$)– [a,b] | $6.15\times10^{-3}$ | 7.61E-87 | 32.2 | -2910 |
| =C–CH< | =CHOO$^\bullet$ | $8.73\times10^{-1}$ | 7.44E-68 | 25.98 | -1906 |
| | =C(OO$^\bullet$)– [a,b] | $8.50\times10^{-3}$ | 7.61E-87 | 32.2 | -3006 |
| **1,6-H-migration** | | | | | |
| =C–CH$_3$ | –CH$_2$OO$^\bullet$ | $3.03\times10^{-2}$ | 4.65E-75 | 27.56 | -3167 |
| | >CHOO$^\bullet$ [a,b] | $1.55\times10^{-2}$ | 3.42E-67 | 25.07 | -1803 |
| | >C(OO$^\bullet$)– [a,b] | $2.99\times10^{-3}$ | 3.42E-67 | 25.07 | -1313 |
| =C–CH$_2$– | –CH$_2$OO$^\bullet$ | $6.50\times10^{-1}$ | 7.21E-70 | 25.88 | -3373 |
| | >CHOO$^\bullet$ [a,b] | $9.46\times10^{-1}$ | 3.42E-67 | 25.07 | -3028 |
| | >C(OO$^\bullet$)– [a,b] | $3.77\times10^{-1}$ | 3.42E-67 | 25.07 | -2754 |
| =C–CH< | –CH$_2$OO$^\bullet$ | $1.37\times10^{1}$ | 1.20E-56 | 21.76 | -2204 |
| | >CHOO$^\bullet$ [a,b] | $8.23\times10^{1}$ | 3.42E-67 | 25.07 | -4359 |
| | >C(OO$^\bullet$)– [a,b] | $2.96\times10^{1}$ | 3.42E-67 | 25.07 | -4054 |
| **1,7-H-migration** | | | | | |
| =C–CH$_3$ | –CH$_2$OO$^\bullet$ | $1.20\times10^{0}$ | 8.54E-60 | 22.36 | -2615 |
| | >CHOO$^\bullet$ [a,b] | $3.72\times10^{-1}$ | 3.95E-52 | 19.88 | -1227 |
| | >C(OO$^\bullet$)– [a,b] | $3.16\times10^{-3}$ | 3.95E-52 | 19.88 | 194 |
| =C–CH$_2$– | –CH$_2$OO$^\bullet$ | $3.28\times10^{0}$ | 1.83E-44 | 17.39 | -836 |
| | >CHOO$^\bullet$ [a,b] | $3.96\times10^{0}$ | 3.95E-52 | 19.88 | -1932 |
| | >C(OO$^\bullet$)– [a,b] | $8.85\times10^{-1}$ | 3.95E-52 | 19.88 | -1486 |
| =C–CH< | –CH$_2$OO$^\bullet$ [a,b] | $3.80\times10^{3}$ | 3.95E-52 | 19.88 | -3978 |
| | >CHOO$^\bullet$ [a,b] | $3.53\times10^{2}$ | 3.95E-52 | 19.88 | -3270 |
| | >C(OO$^\bullet$)– [a,b] | $6.68\times10^{1}$ | 3.95E-52 | 19.88 | -2774 |

[a] Averaged temperature-dependence from other reactions used

[b] k(298 K) anchor value derived from relative rates of aliphatic RO$_2$, scaled to unsaturated RO$_2$



*Table 3: Rate coefficients for allylic H-migration in peroxy radicals with a double bond outside the TS ring, accounting for all abstractable H-atoms in the group.*

| Substitution pattern | | $k(298K)$ / $s^{-1}$ | $k(200\text{-}450K) = A \times (T/K)^n \times \exp(-Ea / T)$ | | |
|---|---|---|---|---|---|
| H-atom | $-OO^\bullet$ | | $A$ / $s^{-1}$ | $n$ | $E_a$ / K |
| **1,4-H-migration** | | | | | |
| $=C-CH_2-$ | $-CH_2OO^\bullet$ | $1.56\times10^{-5}$ | 9.82E+01 | 2.65 | 9172 |
| | $>CHOO^{\bullet\ a,b}$ | $7.72\times10^{-6}$ | 9.82E+01 | 2.65 | 9382 |
| | $>C(OO^\bullet)-^{\ a,b}$ | $2.55\times10^{-5}$ | 9.82E+01 | 2.65 | 9026 |
| $=C-CH<$ | $-CH_2OO^{\bullet\ a}$ | $9.52\times10^{-5}$ | 9.82E+01 | 2.65 | 8634 |
| | $>CHOO^{\bullet\ a,b}$ | $7.50\times10^{-4}$ | 9.82E+01 | 2.65 | 8019 |
| | $>C(OO^\bullet)-^{\ a,b}$ | $8.66\times10^{-3}$ | 9.82E+01 | 2.65 | 7290 |
| **1,5-H-migration** | | | | | |
| $=C-CH_2-$ | $-CH_2OO^\bullet$ | $2.71\times10^{0}$ | 5.52E-47 | 18.56 | -525 |
| | $>CHOO^{\bullet\ a,b}$ | $2.58\times10^{-2}$ | 6.10E-25 | 11.23 | 3540 |
| | $>C(OO^\bullet)-$ | $4.60\times10^{-5}$ | 6.74E-03 | 3.9 | 8104 |
| $=C-CH<$ | $-CH_2OO^{\bullet\ a}$ | $6.15\times10^{-1}$ | 6.10E-25 | 11.23 | 2595 |
| | $>CHOO^{\bullet\ a}$ | $1.87\times10^{0}$ | 6.10E-25 | 11.23 | 2263 |
| | $>C(OO^\bullet)-^{\ a,b}$ | $1.30\times10^{-2}$ | 6.10E-25 | 11.23 | 3744 |
| **1,6-H-migration** | | | | | |
| $=C-CH_2-$ | $-CH_2OO^\bullet$ | $2.04\times10^{0}$ | 8.49E-55 | 20.84 | -1928 |
| | $>CHOO^{\bullet\ a,b}$ | $7.68\times10^{-2}$ | 7.71E-37 | 15.05 | 1531 |
| | $>C(OO^\bullet)-$ | $4.27\times10^{0}$ | 7.72E-46 | 17.86 | -1061 |
| $=C-CH<$ | $-CH_2OO^{\bullet\ a}$ | $7.39\times10^{-1}$ | 7.71E-37 | 15.05 | 857 |
| | $>CHOO^{\bullet\ a,b}$ | $6.69\times10^{0}$ | 7.71E-37 | 15.05 | 200 |
| | $>C(OO^\bullet)-$ | $4.17\times10^{-3}$ | 6.99E-10 | 6.43 | 6275 |
| **1,7-H-migration** | | | | | |
| $=C-CH_2-$ | $-CH_2OO^\bullet$ | $8.22\times10^{0}$ | 4.62E-48 | 18.62 | -1503 |
| | $>CHOO^{\bullet\ a,b}$ | $2.55\times10^{-2}$ | 4.62E-48 | 18.62 | 219 |
| | $>C(OO^\bullet)-^{\ a,b}$ | $5.70\times10^{-3}$ | 4.62E-48 | 18.62 | 665 |
| $=C-CH<$ | $-CH_2OO^{\bullet\ a}$ | $3.56\times10^{-1}$ | 4.62E-48 | 18.62 | -568 |
| | $>CHOO^{\bullet\ a,b}$ | $2.27\times10^{0}$ | 4.62E-48 | 18.62 | -1120 |
| | $>C(OO^\bullet)-^{\ a,b}$ | $4.30\times10^{-1}$ | 4.62E-48 | 18.62 | -624 |

[a] Averaged temperature-dependence from other reactions used

[b] $k(298\ K)$ anchor value derived from relative rates of aliphatic $RO_2$, scaled to unsaturated $RO_2$





*Table 4: Rate coefficients for H-migration in aldehydic peroxy radicals.*

| Substitution pattern | | $k(298K) / s^{-1}$ | $k(200\text{-}450K) = A \times (T/K)^n \times \exp(-E_a / T)$ | | |
|---|---|---|---|---|---|
| H-atom | $-OO^{\bullet}$ | | $A / s^{-1}$ | $n$ | $E_a / K$ |
| | | **1,4-H-migration** | | | |
| $-CH=O$ | $-CH_2OO^{\bullet}$ [a] | $2.56 \times 10^{0}$ | 5.24E-65 | 24.74 | -2379 |
| | $>CHOO^{\bullet}$ | $6.64 \times 10^{-2}$ | 1.08E-66 | 25.23 | -1616 |
| | $>C(OO^{\bullet})-$ | $5.70 \times 10^{-1}$ | 2.54E-63 | 24.25 | -1605 |
| | | **1,5-H-migration** | | | |
| $-CH=O$ | $-CH_2OO^{\bullet}$ | $9.84 \times 10^{0}$ | 3.66E-37 | 15.46 | 573 |
| | $>CHOO^{\bullet}$ [a,b] | $1.62 \times 10^{2}$ | 3.66E-37 | 15.46 | 423 |
| | $>C(OO^{\bullet})-$ [a,b] | $1.74 \times 10^{-1}$ | 3.66E-37 | 15.46 | 1775 |
| | | **1,6-H-migration** | | | |
| $-CH=O$ | $-CH_2OO^{\bullet}$ | $4.98 \times 10^{0}$ | 2.03E-41 | 16.79 | 101 |
| | $>CHOO^{\bullet}$ [a,b] | $790 \times 10^{0}$ | 2.03E-41 | 16.79 | -37 |
| | $>C(OO^{\bullet})-$ [a] | $2.84 \times 10^{-1}$ | 2.03E-41 | 16.79 | 268 |
| | | **1,7-H-migration** | | | |
| $-CH=O$ | $-CH_2OO^{\bullet}$ | $1.84 \times 10^{0}$ | 2.91E-30 | 12.9 | 1458 |
| | $>CHOO^{\bullet}$ [a,b] | $1.71 \times 10^{-1}$ | 2.91E-30 | 12.9 | 2167 |
| | $>C(OO^{\bullet})-$ [a,b] | $3.24 \times 10^{-2}$ | 2.91E-30 | 12.9 | 2663 |

[a] Averaged temperature-dependence from other reactions used

[b] k(298 K) anchor value derived from relative rates of aliphatic $RO_2$, scaled to aldehydic $RO_2$






*Table 5: Rate coefficients corrections for oxo-, hydroxy-, nitrate- or alkoxy-substituted H-atoms in RO$_2$ radicals, or for acylperoxy radicals, relative to H-migration in aliphatic peroxy radicals, accounting for all abstractable H-atoms in the group.*

| H-atom | span | reference rate | correction factor |
|---|---|---|---|
| endo-β-oxo | 1,5 | $k_{aliphatic}(T)$ | exp(-450 K/T) |
| | 1,6 | $k_{aliphatic}(T)$ | exp(650 K/T) |
| | 1,7 | $k_{aliphatic}(T)$ | exp(1200 K/T) |
| exo-β-oxo | 1,4 | $k_{aliphatic}(T)$ | exp(2250 K/T) |
| | 1,5 | $k_{aliphatic}(T)$ | exp(1150 K/T) |
| | 1,6 | $k_{aliphatic}(T)$ | exp(700 K/T) |
| | 1,7 | $k_{aliphatic}(T)$ | exp(550 K/T) |
| β-OH | any | $k_{aliphatic}(T)$ | exp(-530K/T) |
| α-ONO$_2$ | any | $k_{aliphatic}(T)$ or $k_{allylic}(T)$ | exp(-950 K/T) |
| α-OR | any | $k_{α-OH}(T)$ | 1 |
| **OO$^\bullet$-group** | **span** | **reference rate** | **correction factor** |
| C(=O)OO$^\bullet$ | any | $k_{aliphatic}(T)$ | exp(900 K/T) |



*Table 6: Rate coefficients for α-OH-substituted H-migration in peroxy radicals, accounting for all abstractable H-atoms in the group.*

| Substitution pattern | | $k(298K) / s^{-1}$ | $k(200\text{-}450K) = A \times (T/K)^n \times \exp(-Ea / T)$ | | |
|---|---|---|---|---|---|
| H-atom | $-OO^\bullet$ | | $A / s^{-1}$ | n | $E_a / K$ |
| *1,4-H-migration* | | | | | |
| $-CH_2OH$ | $-CH_2OO^{\bullet}$ [a,b] | $9.79\times10^{-4}$ | 8.05E-02 | 3.81 | 7790 |
| | $>CHOO^{\bullet}$ [a] | $2.40\times10^{-6}$ | 8.05E-02 | 3.81 | 9581 |
| | $>C(OO^\bullet)-$ [a,b] | $2.06\times10^{-5}$ | 8.05E-02 | 3.81 | 8941 |
| $-CHOH-$ | $-CH_2OO^{\bullet}$ [a] | $8.25\times10^{-6}$ | 8.05E-02 | 3.81 | 9213 |
| | $>CHOO^{\bullet}$ [a] | $3.13\times10^{-5}$ | 8.05E-02 | 3.81 | 8816 |
| | $>C(OO^\bullet)-$ [a,b] | $6.97\times10^{-3}$ | 8.05E-02 | 3.81 | 7205 |
| *1,5-H-migration* | | | | | |
| $-CH_2OH$ | $-CH_2OO^{\bullet}$ [a,b] | $8.39\times10^{-3}$ | 8.05E-02 | 3.81 | 7150 |
| | $>CHOO^{\bullet}$ [a,b] | $1.27\times10^{-2}$ | 8.05E-02 | 3.81 | 7026 |
| | $>C(OO^\bullet)-$ [a,b] | $4.65\times10^{-3}$ | 8.05E-02 | 3.81 | 7326 |
| $-CHOH-$ | $-CH_2OO^{\bullet}$ | $4.42\times10^{-2}$ | 5.54E-02 | 4.12 | 7059 |
| | $>CHOO^{\bullet}$ [a] | $9.55\times10^{-2}$ | 8.05E-02 | 3.81 | 6425 |
| | $>C(OO^\bullet)-$ [a] | $3.31\times10^{-1}$ | 8.05E-02 | 3.81 | 6055 |
| *1,6-H-migration* | | | | | |
| $-CH_2OH$ | $-CH_2OO^{\bullet}$ [a] | $8.84\times10^{-3}$ | 8.05E-02 | 3.81 | 7134 |
| | $>CHOO^{\bullet}$ | $3.10\times10^{-1}$ | 3.41E+07 | 0.86 | 6972 |
| | $>C(OO^\bullet)-$ [a,b] | $4.58\times10^{-3}$ | 8.05E-02 | 3.81 | 7330 |
| $-CHOH-$ | $-CH_2OO^{\bullet}$ [a] | $1.82\times10^{-1}$ | 8.05E-02 | 3.81 | 6233 |
| | $>CHOO^{\bullet}$ | $1.88\times10^{-1}$ | 3.03E-01 | 3.39 | 5905 |
| | $>C(OO^\bullet)-$ [a,b] | $3.60\times10^{-1}$ | 8.05E-02 | 3.81 | 6030 |
| *1,7-H-migration* | | | | | |
| $-CH_2OH$ | $-CH_2OO^{\bullet}$ [a,b] | $6.86\times10^{-4}$ | 8.05E-02 | 3.81 | 7896 |
| | $>CHOO^{\bullet}$ [a,b] | $1.46\times10^{-4}$ | 8.05E-02 | 3.81 | 8358 |
| | $>C(OO^\bullet)-$ [a,b] | $3.26\times10^{-5}$ | 8.05E-02 | 3.81 | 8804 |
| $-CHOH-$ | $-CH_2OO^{\bullet}$ | $1.40\times10^{-1}$ | 1.17E-01 | 3.51 | 5907 |
| | $>CHOO^{\bullet}$ [a,b] | $1.30\times10^{-2}$ | 8.05E-02 | 3.81 | 7019 |
| | $>C(OO^\bullet)-$ [a,b] | $2.46\times10^{-3}$ | 8.05E-02 | 3.81 | 7515 |

[a] Averaged temperature-dependence from other reactions used

[b] k(298 K) anchor value derived from relative rates of aliphatic $RO_2$, scaled to α-OH $RO_2$





*Table 7: Rate coefficients for H-migration of α-OOH H-atoms in peroxy radicals, accounting for all abstractable H-atoms in the group.*

| Substitution pattern | | $k(298K) / s^{-1}$ | $k(200\text{-}450K) = A \times (T/K)^n \times \exp(-E_a / T)$ | | |
|---|---|---|---|---|---|
| H-atom | $-OO^\bullet$ | | $A / s^{-1}$ | $n$ | $E_a / K$ |
| 1,4-H-migration | | | | | |
| $-CH_2OOH$ | $-CH_2OO^\bullet$ | $2.37\times10^{-8}$ | 1.96E+07 | 1.61 | 12969 |
| | $>CHOO^\bullet$ | $8.77\times10^{-9}$ | 2.38E+04 | 2.69 | 13103 |
| | $>C(OO^\bullet)-$ | $3.56\times10^{-7}$ | 2.04E+05 | 2.45 | 12228 |
| $-CH(OOH)-$ | $-CH_2OO^\bullet$ | $2.33\times10^{-7}$ | 7.83E+04 | 2.24 | 11716 |
| | $>CHOO^\bullet$ | $1.39\times10^{-6}$ | 1.22E+05 | 2.38 | 11559 |
| | $>C(OO^\bullet)-$ | $5.30\times10^{-7}$ | 9.91E+05 | 1.86 | 11587 |
| 1,5-H-migration | | | | | |
| $-CH_2OOH$ | $-CH_2OO^\bullet$ [a] | $4.13\times10^{-4}$ | 2.01E-52 | 19.91 | 654 |
| | $>CHOO^\bullet$ [a] | $5.70\times10^{-4}$ | 2.01E-52 | 19.91 | 558 |
| | $>C(OO^\bullet)-$ [a] | $1.89\times10^{-2}$ | 2.01E-52 | 19.91 | -484 |
| $-CH(OOH)-$ | $-CH_2OO^\bullet$ [a] | $5.99\times10^{-3}$ | 2.01E-52 | 19.91 | -143 |
| | $>CHOO^\bullet$ [a] | $2.18\times10^{-2}$ | 2.01E-52 | 19.91 | -527 |
| | $>C(OO^\bullet)-$ [a] | $2.75\times10^{-2}$ | 2.01E-52 | 19.91 | -597 |
| 1,6-H-migration | | | | | |
| $-CH_2OOH$ | $-CH_2OO^\bullet$ [a] | $3.01\times10^{-3}$ | 2.01E-52 | 19.91 | 62 |
| | $>CHOO^\bullet$ | $1.01\times10^{-3}$ | 3.77E-40 | 15.89 | 1981 |
| | $>C(OO^\bullet)-$ [a] | $8.77\times10^{-4}$ | 2.01E-52 | 19.91 | 430 |
| $-CH(OOH)-$ | $-CH_2OO^\bullet$ | $5.70\times10^{-2}$ | 1.07E-64 | 23.93 | -2406 |
| | $>CHOO^\bullet$ [a] | $3.04\times10^{-2}$ | 2.01E-52 | 19.91 | -627 |
| | $>C(OO^\bullet)-$ [a] | $2.45\times10^{-2}$ | 2.01E-52 | 19.91 | -563 |
| 1,7-H-migration | | | | | |
| $-CH_2OOH$ | $-CH_2OO^\bullet$ [a] | $2.72\times10^{-4}$ | 2.01E-52 | 19.91 | 778 |
| | $>CHOO^\bullet$ [a] | $5.44\times10^{-4}$ | 2.01E-52 | 19.91 | 572 |
| | $>C(OO^\bullet)-$ [a] | $1.90\times10^{-4}$ | 2.01E-52 | 19.91 | 886 |
| $-CH(OOH)-$ | $-CH_2OO^\bullet$ [a] | $1.30\times10^{-2}$ | 2.01E-52 | 19.91 | -374 |
| | $>CHOO^\bullet$ [a] | $1.29\times10^{-2}$ | 2.01E-52 | 19.91 | -371 |
| | $>C(OO^\bullet)-$ [a] | $1.07\times10^{-2}$ | 2.01E-52 | 19.91 | -315 |

[a] Averaged temperature-dependence from other reactions used

[b] k(298 K) anchor value derived from relative rates of aliphatic $RO_2$, scaled to α-OOH $RO_2$





*Table 8: Rate coefficients for H-migration between –OOH groups and peroxy radicals.*

| Substitution pattern | | $k(298K)$ / $s^{-1}$ | $k(200\text{-}450K) = A \times (T/K)^n \times \exp(-E_a / T)$ | | |
|---|---|---|---|---|---|
| H-atom | –OO• | | $A$ / $s^{-1}$ | $n$ | $E_a$ / K |
| **1,6-H-migration** | | | | | |
| –CH$_2$OOH | –CH$_2$OO• [a,b] | $1.90\times10^2$ | 5.64E-20 | 8.46 | -405 |
| | >CHOO• [a,b] | $2.51\times10^1$ | 5.64E-20 | 8.46 | 198 |
| | >C(OO•)– [a,b] | $2.47\times10^2$ | 5.64E-20 | 8.46 | -484 |
| –CH(OOH)– | –CH$_2$OO• [a,b] | $7.05\times10^2$ | 5.64E-20 | 8.46 | -796 |
| | >CHOO• [a,b] | $1.54\times10^3$ | 5.64E-20 | 8.46 | -1029 |
| | >C(OO•)– [a,b] | $1.00\times10^2$ | 5.64E-20 | 8.46 | -214 |
| >C(OOH)– | –CH$_2$OO• [a,b] | $1.10\times10^3$ | 5.64E-20 | 8.46 | -928 |
| | >CHOO• [a,b] | $1.34\times10^4$ | 5.64E-20 | 8.46 | -1674 |
| | >C(OO•)– [a,b] | $2.62\times10^3$ | 5.64E-20 | 8.46 | -1187 |
| **1,7-H-migration** | | | | | |
| –CH$_2$OOH | –CH$_2$OO• [a,b] | $3.26\times10^2$ | 5.64E-20 | 8.46 | -566 |
| | >CHOO• [a,b] | $6.07\times10^2$ | 5.64E-20 | 8.46 | -751 |
| | >C(OO•)– [a,b] | $2.13\times10^3$ | 5.64E-20 | 8.46 | -1126 |
| –CH(OOH)– | –CH$_2$OO• [a,b] | $1.59\times10^3$ | 5.64E-20 | 8.46 | -1038 |
| | >CHOO• [a,b] | $2.09\times10^3$ | 5.64E-20 | 8.46 | -1119 |
| | >C(OO•)– [a,b] | $1.16\times10^4$ | 5.64E-20 | 8.46 | -1630 |
| >C(OOH)– | –CH$_2$OO• [a,b] | $5.14\times10^3$ | 5.64E-20 | 8.46 | -1388 |
| | >CHOO• [a,b] | $9.92\times10^3$ | 5.64E-20 | 8.46 | -1584 |
| | >C(OO•)– [a,b] | $7.84\times10^2$ | 5.64E-20 | 8.46 | -827 |
| **1,8-H-migration** | | | | | |
| –CH$_2$OOH | –CH$_2$OO• [a,b] | $6.42\times10^1$ | 5.64E-20 | 8.46 | -82 |
| | >CHOO• | $1.18\times10^2$ | 9.14E-15 | 6.76 | 423 |
| | >C(OO•)– [a,b] | $1.92\times10^2$ | 5.64E-20 | 8.46 | -408 |
| –CH(OOH)– | –CH$_2$OO• | $4.17\times10^2$ | 3.48E-25 | 10.16 | -1327 |
| | >CHOO• [a,b] | $2.46\times10^2$ | 5.64E-20 | 8.46 | -482 |
| | >C(OO•)– [a,b] | $1.36\times10^3$ | 5.64E-20 | 8.46 | -991 |
| >C(OOH)– | –CH$_2$OO• [a,b] | $2.51\times10^2$ | 5.64E-20 | 8.46 | -489 |
| | >CHOO• [a,b] | $1.58\times10^2$ | 5.64E-20 | 8.46 | -350 |
| | >C(OO•)– [a,b] | $2.25\times10^2$ | 5.64E-20 | 8.46 | -456 |
| **1,9-H-migration** | | | | | |



| –CH$_2$OOH | –CH$_2$OO$^\bullet$ [a,b] | 2.68×10$^0$ | 5.64E-20 | 8.46 | 864 |
|---|---|---|---|---|---|
| | >CHOO$^\bullet$ [a,b] | 3.34×10$^0$ | 5.64E-20 | 8.46 | 791 |
| | >C(OO$^\bullet$)– [a,b] | 8.30×10$^0$ | 5.64E-20 | 8.46 | 528 |
| –CH(OOH)– | –CH$_2$OO$^\bullet$ [a,b] | 3.79×10$^1$ | 5.64E-20 | 8.46 | 75 |
| | >CHOO$^\bullet$ [a,b] | 1.03×10$^1$ | 5.64E-20 | 8.46 | 464 |
| | >C(OO$^\bullet$)– [a,b] | 5.67×10$^1$ | 5.64E-20 | 8.46 | -45 |
| >C(OOH)– | –CH$_2$OO$^\bullet$ [a,b] | 1.09×10$^1$ | 5.64E-20 | 8.46 | 448 |
| | >CHOO$^\bullet$ [a,b] | 6.82×10$^0$ | 5.64E-20 | 8.46 | 586 |
| | >C(OO$^\bullet$)– [a,b] | 9.72×10$^0$ | 5.64E-20 | 8.46 | 481 |

[a] Averaged temperature-dependence from other reactions used

[b] k(298 K) anchor value derived from scaled values (see text)






*Table 9: Comparison of the SAR predictions, $k_{SAR}(T)$ against the experimental values, $k_{exp}(T)$*

| Reference | Molecule | span / subst. | T / K | $k_{exp}(T)$ | $k_{SAR}(T)$ |
|---|---|---|---|---|---|
| Nozière and | •OO-CH$_2$-CH$_2$-CH$_2$-CH$_3$ | all | 298 | $\leq (5 \pm 3) \times 10^{-4}$ | $8.1 \times 10^{-4}$ |
| Vereecken, 2019 | •OO-CH$_2$-CH$_2$-CH$_2$-CH$_2$-CH$_3$ | all | 298 | $(2.4 \pm 1.3) \times 10^{-3}$ | $1.7 \times 10^{-3}$ |
| | •OO-CH$_2$-CH$_2$-CH$_2$-CH$_2$-CH$_2$-CH$_3$ | all | 298 | $(1.7 \pm 0.6) \times 10^{-3}$ | $1.8 \times 10^{-3}$ |
| | •OO-CH$_2$-CH(C$_2$H$_5$)-CH$_2$-CH$_2$-CH$_2$-CH$_3$ | all | 298 | $(2.2 \pm 1.5) \times 10^{-3}$ | $2.5 \times 10^{-3}$ |
| | mixture of 2,5-diMe-hexylperoxy [a] | all | 298 | $(1.9 \pm 0.6) \times 10^{-2}$ | $2.7 \times 10^{-2}$ |
| (Crounse et al., 2012) | HOCH$_2$-C(CH$_3$)(OO•)-CH=O | 1,4 aldehyde-H | 296 | $0.5 \pm 0.3$ | 0.5 [b] |
| (Crounse et al., 2013) | •OO-CH(CH$_3$)-C(=O)-CH$_2$-CH$_3$ | 1,5 endo-β-oxo | 296 | $\leq 2 \times 10^{-3}$ | $2.2 \times 10^{-4}$ |
| | •OO-CH(CH$_3$)-C(=O)-CH(OOH)-CH$_3$ | 1,5 α-OOH + endo-β-oxo | 296 | $> 0.1$ | $4.8 \times 10^{-3}$ |
| (Praske et al., 2018) | CH$_3$-CH$_2$-CH(OO•)-CH$_2$-CH(OH)-CH$_3$ | 1,5 α-OH | 296 | $(4.8 \pm 2.4) \times 10^{-2}$ | $8.0 \times 10^{-2}$ |
| | | | 318 | $(3.1 \pm 1.2) \times 10^{-1}$ | $4.7 \times 10^{-1}$ |
| | CH$_3$-CH(OO•)-CH$_2$-CH$_2$-CH(OH)-CH$_3$ | 1,6 α-OH | 296 | $(1.4 \pm 0.6) \times 10^{-1}$ | $1.6 \times 10^{-1}$ |
| | | | 318 | $(5.5 \pm 1.6) \times 10^{-1}$ | $8.2 \times 10^{-1}$ |
| (Praske et al., 2019) | CH$_3$-C(CH$_3$)(OO•)-CH$_2$-CH(OH)-CH$_3$ | 1,5 α-OH | 296 | $(2.2 \pm 0.9) \times 10^{-1}$ | $8 \times 10^{-2}$ |
| | | | 318 | $(5.6 \pm 2.9) \times 10^{-1}$ | $4.7 \times 10^{-1}$ |
| | CH$_3$-C(CH$_3$)(OO•)-CH$_2$-CH(OOH)-CH$_3$ | 1,5 α-OOH | 296 | $(3.6 \pm 5.2) \times 10^{-2}$ | $2.4 \times 10^{-2}$ |
| | | | 318 | $(1.3 \pm 1.6) \times 10^{-1}$ | $8.8 \times 10^{-2}$ |
| (Heiss and Sahetchian, 1996) | •OO-CH$_2$-CH$_2$-CH$_2$-CH$_2$-OH [c] | 1,6 α-OH [c] | 487 | $1.5 \times 10^{3}$ | $6.1 \times 10^{2}$ |
| (Jorand et al., 2003) | CH$_3$-CH(OO•)-CH$_2$-CH$_2$-CH(OH)-CH$_3$ | 1,6 α-OH | 453 | $(2.5 \pm 0.2) \times 10^{3}$ | $6.9 \times 10^{2}$ |
| | | | 483 | $(6.7 \pm 0.2) \times 10^{3}$ | $1.9 \times 10^{3}$ |
| (Perrin et al., 1998) | CH$_3$-CH(OO•)-CH$_2$-CH$_2$-CH$_2$(OH) | 1,6 α-OH | 463 | $(6.7 \pm 0.6) \times 10^{2}$ | $1.9 \times 10^{3}$ |
| | | | 412 | $(5.5 \pm 0.5) \times 10^{3}$ | $1.2 \times 10^{4}$ |
| | CH$_3$-CH(OO•)-CH$_2$-CH$_2$-CH$_2$-CH$_3$ | 1,5/1,6 [d] | 453 | 0.795 [d] | 0.468 [d] |
| (Teng et al., 2017) | Z-•OO-CH$_2$-CH=C(CH$_3$)-CH$_2$OH | 1,6 endo-allyl | 297 | $0.36 \pm 0.14$ | 0.65 |
| | Z-•OO-CH$_2$-C(CH$_3$)=CH-CH$_2$OH | 1,6 endo-allyl | 297 | $3.7 \pm 1.0$ | 0.65 |

[a] SAR prediction for a mixture of 58:13:29 of 2,5-diMe-hexyl-1-, 2-, and 3-peroxy, respectively, following the procedure in Nozière and Vereecken (2019) for a reaction time of 1 min.

[b] SAR matches experimental value by design. Excluding experimental value from the SAR derivation yields 0.23 s$^{-1}$, showing good agreement between theoretical and experimental data.

[c] The observed reaction is identified based on the product study by (Jorand et al., 1996)

[d] Ratio of the 1,5- over 1,6-H-migration rate.