# Peer review of "H-migration in peroxy radicals under atmospheric conditions"

_Atmospheric Chemistry and Physics, 2019_

## Referee Comment (RC1) · Anonymous Referee #1 · 29 Feb 2020

This paper reports the structure-activity relationship (SAR) for the rate coefficients for the isomerization reactions of organic peroxy radicals (RO2) to QOOH based on the experimental and theoretical investigations. It adds new class of knowledge on the atmospheric chemistry. I request the authors to address the following comment on the assumption in the analysis of experiments.

[1] In the analysis of experimental data, authors simple one-way reaction (R1). RO2 —> HOOQO2 (R1) However, this should be actually the two-step reactions. RO2 —> QOOH (R2) QOOH —> RO2 (R-2) QOOH + O2 —> HOOQO2 (R3) The assumption of simple one-way reaction (R1) is only valid when $k3[O2] \gg k\text{-}2$. My rough calculation shows that $k\text{-}2$ may be as large as $10^7$ to $10^8$ s-1, which is comparable order with $k3[O2]$ at the authors experimental condition, (0.9 atm, 298 K, synthetic air), $k3[O3] =$

3*10ˆ7 $\sim$ 1*10ˆ8. Authors should properly discuss the effects of the backward reactions (R-2).

---

## Referee Comment (RC2) · Anonymous Referee #3 · 20 Mar 2020

This RO2 H-shift SAR is a long-awaited advance in atmospheric chemistry. The authors quite sensibly choose to simplify the enormously complex RO2 chemistry by striving to create a (rather comprehensive) "single-functionality" SAR, which can then (assuming some simple additivity rules) be used as a good initial guess also for more complex multifunctional systems (and e.g. R/S stereochemistry, cyclic structures inside the TS ring, etc). Instead of attempting to generate comprehensive conformer sets for all their structures, they also really heavily on what they call "rel-MC-TST", where the "full" set of conformers (at a given level of quantum chemical theory) is generated only for a reference reaction, and more complex systems are studied by only computing a subset of conformers for those systems, including both ones as similar as possible to the reference system, and others "representing the change in molecular structure

between reference and target systems". While I'm dubious as to the universal applicability of this approach, e.g. for systems with multiple (non-alkyl) functional groups and thus a large number of H-bonding patterns, as in heavily oxidised large RO2, it is likely a good and cost-effective choice for the systems with relatively few non-alkyl functional groups studied here.

The authors are also to be commended for providing their raw data as a supplement - though I'm not sure if the provided data actually contains conformer data as well? (Given how important the authors consider the conformers to be, I would argue that the dataset perhaps SHOULD contain also at least the most important conformers for each system; see also comment 12 above.)

I have a number of somewhat technical issues that I would like to hear the author's comments on. Some of these may require minor revisions to the manuscript, for others I'd just like to hear the authors' response here in the discussion area.

1)Please define exactly what is meant by a "first-order SAR"?

2)Please explain what is meant by the sentence "for the remaining reactions the product energy is assumed to be similar to analogous reactions with similar product characteristics"? I fist assumed this refers to cases where the H-shift leads to prompt decomposition, i.e. where a QOOH "product" minimum does not exist. But upon closer reading, I realised this probably part of the authors' general "rel-MC-TST" philosophy, where computational time is saved by not explicitly computing product energies for all reactions... Anyway, this could if possible be reformulated to make the issue even clearer.

3)The authors argue against H-shifts in "trans"-alkenes by stating "To reduce the strain in the transition states very large migrations spans would then be required, leading again to low rate coefficients due to entropic disadvantage. ". But wouldn't the presence of the double bond remove at least one of the internal rotors, thus reducing the entropic disadvantage? Wouldn't it be worth checking e.g. some 1,8 or 1,9 H-shifts

for trans-alkenes (with otherwise favourable substituents for H-shifts) to make sure the argument holds? Note that similar arguments also apply to conjugated or triple bonds, as well as cyclic substituents within the TS ring, as in many monoterpene-derived RO2: the steric strain likely prevents most H-shifts, but on the other hand the entropic penalty is lesser, potentially making for example even 1,9 H-shifts feasible (e.g. in the d3-carene + O3 system). On the other hand, it's perfectly understandable that the authors wish to limit the scope of their SAR - the number of potential RO2 systems is after all enormous - so the trans-conformers can well be left to a future study.

4)Illustrative (stick diagram) pictures of the different H-shift cases studied could aid readers especially from the "physics-side" of the ACP community. I'm a chemist, but even I needed some time to understand exactly what "allylic H-migration with a endo-cyclic gem-substituted double bond" refers to.

5)The authors report large discrepancies (of unknown origin) between different studies concerning rates of "scrambling" H-shifts between RO2 and ROOH groups. In our experience, transition states for such reactions are particularly prone to artefacts in the Hartree-Fock stage of coupled-cluster calculations, where the HF algorithms used in programs such as Gaussian, Molpro etc. converge to an incorrect, higher-energy solution, analogous to local minima in geometry optimisations. That can then lead to either higher - or sometimes lower, but in any case incorrect - coupled cluster energies. Whether or not this error occurs seems to be almost random, with different HF algorithms (i.e. either different programs, or different algorithm choices within the same program), and different basis sets leading to different results - but sadly none of the combinations we have tested so far (including using the otherwise very robust MCSCF algorithm in Molpro for the initial HF guess) seems to be guaranteed to always give the lowest possible HF energy. For some discussion on this, see for example the Moeller et al 2019 paper cited in the manuscript. One quick way to check if this is an issue (apart from systematically performing "orbital rotation" checks on all HF energies as suggested for example by Vaucher and Reiher, J. Chem. Theory Comput. 2017, 13, 3,

low11219-1228) would be to compare the "DFT-only" level results from different papers on this reaction class. Generally, we would expect (much) larger scatter in the DFT results due to larger inherent errors, and to differences between different functionals, but since DFT seems to be generally less affected than coupled cluster by this "local-minimum orbital" problem, then this case might be the exception. . .

6)Is there a particular reason the authors choose to use canonical CCSD(T), rather than the potentially more cost-effective variants such as CCSD(T)-F12, DLPNO-CCSD(T), or the very recent combination of the two, DLPNO-CCSD(T)-F12? I am fully aware that the details of the CCSD(T) correction is unlikely to be the major error source on the rates, I'm just interested in their reasoning here.

7)One more recent reference containing some "medium-level accuracy" computed RO2 H-shift rates for multifunctional systems (e.g. -ONO2 and C=O; also a C3 ring inside the TS) is: D. Draper et al., Formation of Highly Oxidized Molecules from NO3 Radical Initiated Oxidation of Δ-3-Carene: A Mechanistic Study, ACS Earth Space Chem. 2019, 3, 8, 1460-1470. The authors might wish to compare their SAR predictions also to the rates given therein.

8)In their explanation of rel-MC-TST (supplementary material), I believe they slightly mischaracterise the approach proposed by Moeller et al 2016 (and related approaches): they do not "generate a limited number of conformers at a very low level of theory rather than attempting a near-exhaustive characterization of the full population", instead they explicitly DO attempt to generate the full population at a very low level of theory, and then attempt to come up with cost-effective approaches of limiting the number of calculations needed at higher levels. The authors may well be correct that with currently available method combination s, this then fails to "reliably generate the most contributing conformer" - but that is a different issue. Also, I would note that while, as said, the authors' present approach is likely very appropriate for their present systems, then as chemical complexity increases, something like the Moeller 2016 approach will inevitably be needed. Consider for example the case of a RO2 from a

C10 monoterpene containing 8 or so O atoms - corresponding to the least condensing "monomer" products in monoterpene oxidation. This could easily (depending of course on the number of remaining cyclic structures) contain on the order of 15 or so relevant dihedral angles, implying at least millions of relevant conformers. No matter how clever the selection of "reference" systems, at some point the "judicious selection of conformers by hand" just becomes impossible, and a systematic and automated approach - inevitably relying on analysing a large number of conformers at a lower level, chosen to maximise error cancellation with the desired final level - will be required.

9)The authors could briefly explain what procedure they carry out to ensure they have indeed found the "full" conformer population for their reference systems in rel-MC-TST: presumably this involves scanning over all dihedral angles (in 120-degree increments, or perhaps even smaller?) and then optimising all the obtained combinations (excluding obviously impossible ones) with the desired final DFT method? And then they must have some approach for distinguishing between (and eliminating) duplicates, as multiple inputs can lead to the same DFT local minimum, but with small differences due to the convergence criteria?

10)Line 340: comapred is a typo.

11)While I don't dispute that the SAR performs well, I wonder slightly at how impressive the factor of 2 agreement quoted in the abstract is, given that the total number of experimental data points included by the authors appears to be 22, and the SAR contains a fairly large number of parameters. (I understand the authors do not actually claim it is particularly impressive on its own, I would just like to hear their thoughts on this.)

12)Do the authors have some guidelines on what to do for more complex systems not covered directly by the SAR? Obviously, the SAR prediction is a very good starting point and sanity check - if it's very low, then likely the H-shift will not be relevant anyway, but what about the cases where the SAR prediction is fairly high, and we need a more

precise value? Would the next step be to find the closest reference system covered by the SAR, and attempt to apply the authors rel-MC-TST approach? (If yes, then it would DEFINITELY be good if the authors uploaded also their conformer data - as said I'm not sure from a quick glance if that's already included in the supplementary material).

---

## Author Comment (AC1) · 24 Mar 2020

An in-depth answer to the comment by Referee #1 required a sizable addition to the manuscript, so we already provide a version of this addition at this time, in case any of the referees would like to comment before we submit our full revised version. The text below may still be changed somewhat to allow better integration in the text.

**Comment by Referee #1:**
In the analysis of experimental data, authors simple one-way reaction (R1). RO2→ HOOQO2 (R1) However, this should be actually the two-step reactions. RO2 → QOOH (R2) QOOH → RO2 (R-2) QOOH + O2 → HOOQO2 (R3) The assumption of simple one-way reaction (R1) is only valid when k3[O2] » k-2. My rough calculation shows that k-2 may be as large as 10^7 to 10^8 s-1, which is comparable order with k3[O2] at the authors experimental condition, (0.9 atm, 298 K, synthetic air), k3[O3] =3*10^7∼1*10^8. Authors should properly discuss the effects of the backward reactions (R-2)

**Answer**:

The reversibility of the reaction is certainly an interesting aspect of the chemistry. We have collected our thoughts in a new section we propose to include in the paper, where we find that the available data suggests that reverse H-migration is not expected to be a dominant path for the alkyl radical products. Although we were aware that the H-migration (R1) could be somewhat reversible, the experimental analysis only takes reaction (R1) into account as a simplification for the three reactions (R1)-(R-2)-(R3). The rate coefficient measured experimentally, $k_{iso}$, is the effective rate coefficient for product formation, as governed by the competition between R-2 and R3, i.e. we only observe the overall phenomenological rate coefficient. The reverse reaction (R-2) appears to be significantly slower than (R3) by an order of magnitude or more, and certainly can not be a dominant path, otherwise the net isomerization could not have been observed experimentally, and could not have had the same good agreement with the theoretical data. As (R3) is much faster than (R1), the later was the kinetically limiting step in the reaction system (R1)-(R-2)-(R3), and the effective rate for product formation was reported as the rate of H-migration. The text will be changed at the appropriate places to reflect this simplification, referring to the new section below.

**New section: The fate of the radical products**

In atmospheric chemistry, the unimolecular reactions of an alkyl radical are typically not considered explicitly. Mechanisms such as the Master Chemical Mechanism assume, by default, $O_2$ addition for all alkyl radicals (Jenkin et al., 1997), unless a special reaction is known to occur; examples of the latter include >C•(OH) + $O_2$, epoxide formation in C•-COOH,  or OH elimination in C•OOH. For aliphatic alkyl radicals, the rate of H-migration or elimination reactions rarely reaches $10^4$ s$^{-1}$ at room temperature, such that $O_2$ addition is dominant (Miyoshi, 2011; Villano et al., 2011; Wang et al., 2015). The weaker OO–H bond in hydroperoxides, however, facilitates H-migration in •QOOH radicals, reversing the $RO_2$ H-migration. If we assume that only the reverse •QOOH to $RO_2$ reaction can occur in competition with $O_2$ addition, the effective rate of $HOOQO_2$ formation will be dependent on the rate of •QOOH formation, $k_1$, and of the reverse reaction $k_{-1}$ competing against the $O_2$ addition on the alkyl radical site, $k_{O2}[O_2]$:

$$k_{eff}(T)=k_1(T)\frac{k_{-1}(T)}{k_{-1}(T)+k_{O2}(T)[O_2]}$$

[revised manuscript text omitted]

For non-alkyl radical products, reversal of the reaction may be more important. For oxygen-based product radicals this has been discussed in detail, see the sections on hydroxy-, hydroperoxy-, carboxylic H-migration.

---

## Referee Comment (RC3) · Anonymous Referee #2 · 31 Mar 2020

The authors present an extensive structure activity relationship for the prediction of temperature-dependent rate coefficients for H-shift isomerisation reactions in substituted alkyl peroxy radicals under atmospheric conditions. Training data for the construction of the SAR is mainly based on (high level) theoretical calculations, supplemented with some experimental data. Both literature theoretical and experimental datasets are used to evaluate the SAR, including some product studies reported in this work.

This is an excellent, well written and thorough piece of work, providing much needed chemical insight into unimolecular $RO_2$ H-migration reactions. Such theoretical SARs, anchored to experimental studies, are crucial in developing our chemical understanding of atmospheric chemistry, providing focus for targeting research in areas of particular uncertainty and in deriving chemical mechanism for scientific and policy modelling. The development of the SAR is described well, including limitations, simplifications and thorough descriptions of the theoretical techniques applied. The SAR is presented in a form that is easy to read and implement into automatic mechanism development methodologies. There is a very useful and comprehensive supplementary folder supplied.

I recommend publication into ACP after the following suggestions and clarifications are considered and addressed:

(1) In terms of chemical notation, I would recommend that IUPAC notation is followed through out. All rate expression terms given in italics e.g. $k_{298\ K}$ or $k(298\ K)$., $k(T)$ , $A$, etc…

It would also be useful to clarify and define the way you are using the modified form of the Arrhenius expression (Kooij formula) in the main text and the SAR tables. Why is $(T/K)^n$ used and not $(T)^n$ ?

(2) This work seems to nicely follow on from discussions in Jenkin et al., (2019) [https://doi.org/10.5194/acp-19-7691-2019] on their SAR for "Estimation of rate coefficients and branching ratios for reactions of organic peroxy radicals" where they introduce discussions on the need for systematic structure–activity methods for a wide range of $RO_2$ radicals and their potential isomerization reactions. They also discuss that the rates of the reverse isomerization reactions are sometimes sufficiently rapid that the product radical may not be fully trapped by onward reaction (e.g. addition of O2) under atmospheric conditions" The way the SAR in this work is set up is that the reverse reactions are not explicitly taken into account (for good reasons, given the theoretical complexities involved). More discussion on this is needed but I see that one of the Referees has brought this up and the authors have already responded.

(3) It would be useful to the reader if more schematic figures and worked examples were shown in order to show exactly how the SARs work/ can be implemented and to illustrate how some of the more interesting reactions work

e.g. the effects of allyl resonance stability in unsaturated $RO_2$ species. It is also unclear to me how the product studies work really links into the evaluation of the SAR. Here, showing how the reaction mechanism for *n*-pentyl $RO_2$ is predicted by the SAR in Figure 10 would be useful. It would also be useful to add the structures associated with the masses show in Figure 9.

(4) Test vs. Training dataset. It would be useful to highlight your strategy of picking what (Scarce) experimental data was used in the training vs. test datasets early on in the manuscript when talking about the development of the SARs. Some of this is brought out more in the evaluation section. It would also be useful to distill out the experimental dataset used into a fully referenced Table in the supplementary.

---

## Author Response (AR1)

**Response to Reviews**

We thank the referees for their helpful comments, which allowed us to improve the manuscript significantly. Some of the questions raised have been addressed only in this response, as the discussion is too specialized for inclusion in the current paper.

**Anonymous Referee #1**

This paper reports the structure-activity relationship (SAR) for the rate coefficients for the isomerization reactions of organic peroxy radicals (RO2) to QOOH based on the experimental and theoretical investigations. It adds new class of knowledge on the atmospheric chemistry. I request the authors to address the following comment on the assumption in the analysis of experiments.[1] In the analysis of experimental data, authors simple one-way reaction (R1). RO2—> HOOQO2 (R1) However, this should be actually the two-step reactions. RO2 —>QOOH (R2) QOOH —> RO2 (R-2) QOOH + O2 —> HOOQO2 (R3) The assumption of simple one-way reaction (R1) is only valid when k3[O2] » k-2. My rough calculation shows that k-2 may be as large as 10ˆ7 to 10ˆ8 s-1, which is comparable order withk3[O2] at the authors experimental condition, (0.9 atm, 298 K, synthetic air), k3[O3] =3*10ˆ7∼1*10ˆ8. Authors should properly discuss the effects of the backward reactions(R-2).

We added a section dedicated to the reverse reaction; as this is a significant extension of the text a draft version was uploaded prior to the end of the reviewing period so referees could get a feel for what we had in mind, and could comment on it.
In this section, we conclude, based on the available literature data, that in near-atmospheric conditions the reverse reaction for (substituted) aliphatic hydrogens is unlikely to affect the rate coefficients by more than about 10%, less than other sources of uncertainty affect the predictions and measurements. As such, both the experimentally observed rates and the rate coefficients included in the SAR are expected to be reliable within the stated uncertainties. For non-aliphatic hydrogens, e.g. for hydroxy-H migration, it is emphasized that the reverse reaction is sometimes expected to be dominant.

**Anonymous Referee #2**

The authors present an extensive structure activity relationship for the prediction of temperature-dependent rate coefficients for H-shift isomerisation reactions in substituted alkyl peroxy radicals under atmospheric conditions. Training data for the construction of the SAR is mainly based on (high level) theoretical calculations, supplemented with some experimental data. Both literature theoretical and experimental datasets are used to evaluate the SAR, including some product studies reported in this work.

40   This is an excellent, well written and thorough piece of work, providing much needed
     chemical insight into unimolecular RO2 H-migration reactions. Such theoretical
     SARs, anchored to experimental studies, are crucial in developing our chemical
     understanding of atmospheric chemistry, providing focus for targeting research in
     areas of particular uncertainty and in deriving chemical mechanism for scientific and
45   policy modelling. The development of the SAR is described well, including
     limitations, simplifications and thorough descriptions of the theoretical techniques
     applied. The SAR is presented in a form that is easy to read and implement into
     automatic mechanism development methodologies. There is a very useful and
     comprehensive supplementary folder supplied.
50   I recommend publication into ACP after the following suggestions and clarifications
     are considered and addressed:

     (1) In terms of chemical notation, I would recommend that IUPAC notation is
     followed through out. All rate expression terms given in italics e.g. k298 K or
55   k(298 K)., k(T) , A, etc...

     We have gone over the text in detail, changing the formatting and naming as requested.

     It would also be useful to clarify and define the way you are using the modified
60   form of the Arrhenius expression (Kooij formula) in the main text and the SAR
     tables. Why is (T/K)n used and not (T)n ?

     The T power factor must be a unit-less value, and (T)n (with n a real number) strictly speaking isn't due
     to the temperature unit of T. In atmospheric chemistry literature it is implicitly understood that T here
65   should be the value of the temperature in Kelvin (dropping the unit), and not e.g. the temperature in °C.
     The $(T/K)^n$ notation merely makes this explicit, forcing a value in Kelvin by unit conversion. We hope
     this helps prevent potential erroneous implementation in software by non-chemists, or complications in
     software that uses physical values (i.e. value + unit objects).

70   (2) This work seems to nicely follow on from discussions in Jenkin et al., (2019)
     [https://doi.org/10.5194/acp-19-7691-2019] on their SAR for "Estimation of
     rate coefficients and branching ratios for reactions of organic peroxy radicals"
     where they introduce discussions on the need for systematic structure–activity
     methods for a wide range of RO2 radicals and their potential isomerization
75   reactions. They also discuss that the rates of the reverse isomerization
     reactions are sometimes sufficiently rapid that the product radical may not be
     fully trapped by onward reaction (e.g. addition of O2) under atmospheric
     conditions" The way the SAR in this work is set up is that the reverse

reactions are not explicitly taken into account (for good reasons, given the
theoretical complexities involved). More discussion on this is needed but I
see that one of the Referees has brought this up and the authors have
already responded.

We now added a new section on the reverse H-migrations. Note that the reverse reaction problem can
never be included in the forward reaction SAR, as there is no guarantee that any reaction from the
product radical is the reverse of the forming reaction. Rather, a separate H-migration SAR for alkyl
radical, alkoxy radicals, and other products must be developed; some such SARs exist in the literature
(e.g. from combustion literature for alkyl radicals, or our earlier work on alkoxy radicals), but
reviewing or developing such SARs is well outside the scope of this paper.

(3) It would be useful to the reader if more schematic figures and worked
examples were shown in order to show exactly how the SARs work/ can be
implemented and to illustrate how some of the more interesting reactions work
e.g. the effects of allyl resonance stability in unsaturated RO2 species.

The supporting information now has examples of 13 reactions spanning a wide range of substituents,
showing the application of the SAR for the derivation of partial and total rate coefficients.

It is also unclear to me how the product studies work really links into the
evaluation of the SAR. Here, showing how the reaction mechanism for n-
pentyl RO2 is predicted by the SAR in Figure 10 would be useful. It would
also be useful to add the structures associated with the masses show in
Figure 9.

The product study section is not part of the section of SAR validation, as it does not derive rate
coefficients. Its purpose is to show that the observed products match the mechanistic details covered by
the SAR, indicating that the reactions proceed indeed following the proposed rearrangements, thus
yielding the anticipated intermediary products. Not all experimental data provides such information,
but track the reaction rate either by loss of reactants, or indirectly by secondary products. We felt that
this information was valuable in this context.
The supporting SAR now has extensive examples on how to apply the SAR, including for aliphatic
cases such as these in figure 9/10.
The peaks figure 9 are now all labeled with a short-notation formula, and the caption explicitly refers to
figure 10 for the Lewis structures.

(4) Test vs. Training dataset. It would be useful to highlight your strategy of
picking what (Scarce) experimental data was used in the training vs. test

datasets early on in the manuscript when talking about the development of the
SARs. Some of this is brought out more in the evaluation section. It would
also be useful to distill out the experimental dataset used into a fully
referenced Table in the supplementary.

Only 4 experimental data points were used in the derivation of the SAR, and 3 of those only by being
included in the geometric averaging across several literature or calculated values to obtain the target
k(298 K). The experimental values used thus are now explicitly marked in Table 9 with an asterisk.
The SAR design section now also explicitly refers to these marked values, explaining how these were
included in the SAR derivation. As only 4 values were used, we feel an additional table is not
warranted; Table 9 is fully referenced. The introduction shortly discusses why not *all* experimental data
can be directly used in the SAR derivation.

Note that the table with experimental data now includes two more items for validation, i.e. one rate and
one rate ratio.

**Anonymous Referee #3**

This RO2 H-shift SAR is a long-awaited advance in atmospheric chemistry. The authors quite sensibly
choose to simplify the enormously complex RO2 chemistry by striving to create a (rather
comprehensive) "single-functionality" SAR, which can then(assuming some simple additivity rules) be
used as a good initial guess also for more complex multi-functional systems (and e.g. R/S
stereochemistry, cyclic structures inside the TS ring, etc). Instead of attempting to generate
comprehensive conformer sets for all their structures, they also really heavily on what they call "rel-
MC-TST", where the "full" set of conformers (at a given level of quantum chemical theory) is
generated only for a reference reaction, and more complex systems are studied by only computing a
subset of conformers for those systems, including both ones as similar as possible to the reference
system, and others "representing the change in molecular structure between reference and target
systems". While I'm dubious as to the universal applicability of this approach, e.g. for systems with
multiple (non-alkyl) functional groups and thus a large number of H-bonding patterns, as in heavily
oxidised large RO2, it is likely a good and cost-effective choice for the systems with relatively few
non-alkyl functional groups studied here.

rel-MC-TST is a relative method, and like relative methods in experimental studies, they work best if
the new systems are as similar as possible to the reference system. Though mathematically rel-MC-
TST will give the correct result even if in cases of poor comparability (i.e. a "universal application" is
possible), it will likely only converge in the limit of many/most/all conformers included in the rate ratio
calculations, negating the benefit of the relative method. Still, even in the case of heavily oxidized large

RO2, rel-MC-TST can be applied with success, but best by using another heavily oxidised large RO2 as reference reaction, comparable to the target system. This point will be emphasized in our separate publication on rel-MC-TST. For now, the text in the SI has an additional paragraph indicating that rel-MC-TST is expected to work best for comparable-sized molecules, that a suitable MC-TST calculation of a reference reaction must be available, and that drastically changing the number of conformers between reference and target reaction makes efficient application hard.

The authors are also to be commended for providing their raw data as a supplement- though I'm not sure if the provided data actually contains conformer data as well? (Given how important the authors consider the conformers to be, I would argue that the dataset perhaps SHOULD contain also at least the most important conformers for each system; see also comment 12 above.)

The supporting information contains the conformer data; for some reactions it was already included in our previous publications (Nozière and Vereecken 2019, Novelli et al. 2020, Fuchs et al. 2018,...).

I have a number of somewhat technical issues that I would like to hear the author's comments on. Some of these may require minor revisions to the manuscript, for others I'd just like to hear the authors' response here in the discussion area.

1) Please define exactly what is meant by a "first-order SAR"?

We now state explicitly in the introduction why we label this a "first-order" SAR:
"*We label this a first-order SAR, as we derive this SAR without rigorously incorporating all literature data available, by including lower-level theoretical predictions, and where needed build on reactivity trends of analogous classes without necessarily being able to validate the transferability of these trends. The current SAR is then a starting framework upon which to build a more complete and accurate SAR as more data becomes available, while retaining immediate usability within the expected accuracy.*"

2) Please explain what is meant by the sentence "for the remaining reactions the product energy is assumed to be similar to analogous reactions with similar product characteristics"? I fist assumed this refers to cases where the H-shift leads to prompt decomposition, i.e. where a QOOH "product" minimum does not exist. But upon closer reading, I realised this probably part of the authors' general "rel-MC-TST" philosophy, where computational time is saved by not explicitly computing product energies for all reactions...Anyway, this could if possible be reformulated to make the issue even clearer.

rel-MC-TST is only used in a subset of the reactions, whereas most calculations used in the SAR as obtained in this paper and the literature are absolute rate determinations; rel-MC-TST is thus not the general philosophy of the SAR.

The omission of explicitly calculating all of the product energy is indeed to reduce the computational cost, as the energy difference between the peroxy radical reactant and the alkyl radical product is well understood based on available bond strengths, and little is gained from calculating dozens of primary, secondary and tertiary alkyl radicals explicitly, especially as the tunneling correction is not sensitive to minor differences between such individual radicals. The text was rephrased to:

"f*or most H-migrations, the reaction energy used in this tunneling correction is estimated from an explicit product characterization (usually 1 conformer only). For the remaining reactions, the product energy is instead estimated based on the product energy of analogous reactions; as the energy of alkyl radicals is well understood, and as the tunneling correction is not overly sensitive to this parameter, this reduced the computational effort without significantly influencing the rate predictions*."

3) The authors argue against H-shifts in "trans"-alkenes by stating "To reduce the strain in the transition states very large migrations spans would then be required, leading again to low rate coefficients due to entropic disadvantage. ". But wouldn't the presence of the double bond remove at least one of the internal rotors, thus reducing the entropic disadvantage? Wouldn't it be worth checking e.g. some 1,8 or 1,9 H-shifts trans-alkenes (with otherwise favourable substituents for H-shifts) to make sure the argument holds? Note that similar arguments also apply to conjugated or triple bonds, as well as cyclic substituents within the TS ring, as in many monoterpene-derived RO2: the steric strain likely prevents most H-shifts, but on the other hand the entropic penalty is lesser, potentially making for example even 1,9 H-shifts feasible (e.g. in the d3-carene + O3 system). On the other hand, it's perfectly understandable that the authors wish to limit the scope of their SAR - the number of potential RO2 systems is after all enormous - so the trans-conformers can well be left to a future study.

We have significantly extended our discussion of migration across unsaturated moieties, and now include triple bonds, trans-alkenes, conjugated double bonds, allenes, and cumulenes. We now refer explicitly to literature for smallest isolable rings containing such unsaturated groups, and make explicit geometric arguments as to the lowest number of backbone atoms we expect. We feel that 1,8-H-shift remain very unlikely, and even 1,9-H-shifts are still likely to have several kcal/mol ring strain, making any additional calculations on such shifts of less importance at this time.
We have not included cyclic structure due to the large number of possible variations and the rather specific impact of each (multi)ring system on the barrier.

4) Illustrative (stick diagram) pictures of the different H-shift cases studied could aid readers especially from the "physics-side" of the ACP community. I'm a chemist, but even I needed some time to understand exactly what "allylic H-migration with a endo-cyclic gem-substituted double bond" refers to.

Each section now has an inset stick diagram of the pertaining reaction, including a reference to the related SAR table, if any. We opted for insets rather than a single large figure to aid the reader in easily finding the section in the text for a reaction of interest.

5) The authors report large discrepancies (of unknown origin) between different studies concerning rates of "scrambling" H-shifts between RO2 and ROOH groups.

We have extended the discussion on the discrepancy for scrambling reactions based on test calculations at the same level of theory as the deviating literature results. We found that B3LYP, used in these latter calculations, has an anomalously large difference, 5 kcal/mol, in the barrier height compared to higher levels of theory, whereas for other H-migrations it is only incorrect by 1-2 kcal/mol (a well-known defect of B3LYP for H-migrations). This is likely affecting the TS geometries which, combined with the fact that no full analysis of the conformational space was done for these larger H-migration spans, is then the likely cause of the mentioned discrepancy.

In our experience, transition states for such reactions are particularly prone to artefacts in the Hartree-Fock stage of coupled-cluster calculations, where the HF algorithms used in programs such as Gaussian, Molpro etc. converge to an incorrect, higher-energy solution, analogous to local minima in geometry optimisations. That can then lead to either higher - or sometimes lower, but in any case incorrect - coupled cluster energies. Whether or not this error occurs seems to be almost random, with different HF algorithms (i.e. either different programs, or different algorithm choices within the same program), and different basis sets leading to different results - but sadly none of the combinations we have tested so far (including using the otherwise very robust MCSCF algorithm in Molpro for the initial HF guess) seems to be guaranteed to always give the lowest possible HF energy. For some discussion on this, see for example the Moeller et al 2019 paper cited in the manuscript. One quick way to check if this is an issue (apart from systematically performing "orbital rotation" checks on all HF energies as suggested for example by Vaucher and Reiher, J. Chem. Theory Comput. 2017, 13, 3, 1219-1228) would be to compare the "DFT-only" level results from different papers on this reaction class. Generally, we would expect (much) larger scatter in the DFT results due to larger inherent errors, and to differences between different functionals, but since DFT seems to be generally less affected than coupled cluster by this "local-minimum orbital" problem, then this case might be the exception...

We are aware of such convergence problems, and routinely compare our DFT results against the CCSD(T) results, and against test calculations at other levels of theory. See for example Vereecken et al. 2017 where we explicitly provide DFT/CCSD(T) comparisons in the supporting information (table 11, p. 29); scatter on the correlation there is a few tenths of a kcal/mol. Another metric for the above convergence failures is that the T1 diagnostic is often, but not always, much higher, even for accompanying CCSD(T) energies close to the expected value. Any metric that deviates from the expected norm is scrutinized in detail, with additional calculations at different basis sets, different

convergence algorithms, and different methodologies, until we are satisfied the result is robust. In SAR development, detecting aberrant calculations is more straightforward than in singular PES studies, as more reference material is available.

Vereecken, L., Novelli, A. and Taraborrelli, D.: Unimolecular decay strongly limits concentration of Criegee intermediates in the atmosphere, Phys. Chem. Chem. Phys., 19, 31599–31612, doi:10.1039/C7CP05541B, 2017.

6) Is there a particular reason the authors choose to use canonical CCSD(T), rather than the potentially more cost-effective variants such as CCSD(T)-F12, DLPNO-CCSD(T), or the very recent combination of the two, DLPNO-CCSD(T)-F12? I am fully aware that the details of the CCSD(T) correction is unlikely to be the major error source on the rates, I'm just interested in their reasoning here.

The listed methods would have been suitable alternatives, but we chose CCSD(T) for practical reasons. We do not have access to a CCSD(T)-F12 program that is applicable to all our projects (e.g. singlet biradicals), so have standardized mostly on CCSD(T) that is available in more software packages. We likewise do not have the computational resources to do CCSD(T)-F12/aug-cc-pVTZ calculations for all species of interest, and with only access to aug-cc-pVDZ or smaller we would not  have the same abilities to verify the behavior of the calculations with respect to systematic basis set series in case of problematic results (see also previous comment), even when we recognize CCSD(T)-F12/aVDZ as being as close or closer to the CBS limit than CCSD(T)/aVTZ. Other practical considerations are the flexibility of the software with regards to (minimum) per-process and overall memory and disk use, the number of software packages to license and keep up to date on our cluster, etc.
The current results are based on calculations spread out over more than a decade, and the above methods were not available to us through the entirety period of time. Using multiple methods within the same SAR is to be avoided, as each method has its own biases and scatter that make it harder to discern trends. This is also one of the reasons why we do not include all of our calculations in the SAR derivation but only those at a few selected levels of theory.

7) One more recent reference containing some "medium-level accuracy" computed RO2H-shift rates for multifunctional systems (e.g. -ONO2 and C=O; also a C3 ring inside the TS) is: D. Draper et al., Formation of Highly Oxidized Molecules from NO3 Radical Initiated Oxidation ofΔ-3-Carene: A Mechanistic Study, ACS Earth Space Chem.2019, 3, 8, 1460-1470. The authors might wish to compare their SAR predictions also to the rates given therein.

As explained in the text, giving an exhaustive overview of the RO2 literature is beyond the scope of this paper, as hundreds of papers would have to be taken into account. At this stage, discussing a handful of additional RO2 radicals with a very specific substructure is unlikely to benefit the overall

goal of our first-order SAR, and instead we intend to use the myriad of literature data for a later, more complete validation and optimization of the SAR.

8) In their explanation of rel-MC-TST (supplementary material), I believe they slightly mis-
characterise the approach proposed by Moeller et al 2016 (and related approaches): they do not "generate a limited number of conformers at a very low level of theory rather than attempting a near-exhaustive characterization of the full population", instead they explicitly DO attempt to generate the full population at a very low level of theory, and then attempt to come up with cost-effective approaches of limiting the number of calculations needed at higher levels. The authors may well be correct that with currently available method combinations, this then fails to "reliably generate the most contributing conformer" - but that is a different issue.
Also, I would note that while, as said, the authors' present approach is likely very appropriate for their present systems, then as chemical complexity increases, something like the Moeller 2016 approach will inevitably be needed. Consider for example the case of a RO2 from a monoterpene containing 8 or so O atoms - corresponding to the least condensing "monomer" products in monoterpene oxidation. This could easily (depending of course on the number of remaining cyclic structures) contain on the order of 15 or so relevant dihedral angles, implying at least millions of relevant conformers. No matter how clever the selection of "reference" systems, at some point the "judicious selection of conformers by hand" just becomes impossible, and a systematic and automated approach -inevitably relying on analysing a large number of conformers at a lower level, chosen to maximise error cancellation with the desired final level - will be required.

To avoid any misrepresentation of the Møller et al. 2016 approach, the text was modified to indicate that "a limited number" is about half of the population, and now specifies that the difference lies in the use of semi-empirical methods versus wavefunction/DFT methods in the sampling. A paragraph was also added indicating that rel-MC-TST is expected to work best for comparable-sized molecules, that a suitable MC-TST calculation of a reference reaction must be available, and that drastically changing the number of conformers between reference and target reaction hampers efficient application. The method is compared to rel-MC-TST specifically because both methods are based on a selection of conformers, and we felt it was necessary to inform less specialized readers on the specific differences in the selection process.

Stating that the Møller et al. 2016 approach explicitly tries to cover all conformers is to our opinion not fully correct, as the semi-empirical methodology is known to miss 50-70% of the reactant and TS conformers, in every benchmark we are aware of (e.g. Møller et al 2016, Novelli et al. 2020). It also misses a similar fraction of the low-energy conformers in complex cases. The Novelli et al. study for example shows that for diHPCARP-RO2, the Møller et al 2016 methodology recovers 600 out of the 1500 reactant conformers (40%) and 8 out of 27 (30%) in the lowest 2 kcal/mol bracket (+3 found by expensive IRC calculations (now 40%)), while dozens of conformers are needed to describe 80 to 90% of the population. It appears that typically about half or more of the population-relevant conformers are not recovered by semi-empirical methods. Similar

conclusions were reached by Vereecken and Peeters in 2002-2003, who therefore chose not to use semi-empirical methods as their lowest-level methodology even though the computational savings would have been very welcome in those days. These details can not be suitable discussed in the current manuscript, but the discussion in Novelli et al 2020 is referenced in the text.

355

We wish to emphasize that we are not opposed to the use of the Møller et al. 2016 technique, and use a lot of data derived with this technique in the SAR derivation. The Møller et al. approach, like full MC-TST and rel-MC-TST, has its place in the spectrum of theoretical kinetic methodologies, all somewhere between full-CI dynamics on diatomics and semi-empirical studies of polymers or proteins. The
360     reduced computational burden of the conformer sampling in Møller et al. 2016 makes it indeed suitable for molecules with a higher number of internal rotors, and we agree with Reviewer 3 that, as chemical complexity increases, some trade-off is inevitably needed to alleviate the computational burden. All of the MC-TST, rel-MC-TST,  Møller et al. 2016 and related approaches have their role to play, each with their own benefits and weaknesses, but the current paper is not the place for a review on kinetic
365     methodologies. The topic was only touched through the necessary but preliminary introduction of rel-MC-TST (see above).

We hope that the changes in the text clarify these points, without going too far into a methodological comparison.

370

9) The authors could briefly explain what procedure they carry out to ensure they have indeed found the "full" conformer population for their reference systems in rel-MC-TST: presumably this involves scanning over all dihedral angles (in 120-degree increments, or perhaps even smaller?) and then optimising all the obtained combinations (excluding obviously impossible ones) with the desired final
375     DFT method? And then they must have some approach for distinguishing between (and eliminating) duplicates, as multiple inputs can lead to the same DFT local minimum, but with small differences due to the convergence criteria?

Referee 3's description is exactly right: we use a template to generate scans over all dihedral angles, in
380     a degree increment relevant for the angle considered (usually 120 degrees as mentioned, smaller for e.g. substituents on rings, or across sp2-sp3 bonds). Good starting angles are readily obtained from general chemical knowledge or simple test calculations.  All geometries are optimized and compared. Detection of doubles compares bond lengths, angles and dihedral angles within a preset tolerance, typically about 0.01 angs for bond lengths, a degree for bond angles and 2-3 degrees for dihedral
385     angles. Convergence to an incorrect molecule is detected by comparison of the molecular graph to a target graph, or visualizing conformers directly.
We do not feel the need to provide so much detail in the paper as other authors are likely to use a very similar approach, as there are no obvious other ones, the differences in the implementations being only in how automated the entire process is. There is also some software available on the web, sometimes
390     with a GUI, for doing such systematic calculations.

Typically we prefer a rate prediction as accurate as feasible, thus aiming to include as many conformers as possible. Some of the aforementioned comparison and culling  is automated but unclear cases are intentionally left to individual decision. Manual checks include metrics such as visualizing the aligned conformers in a chemical viewer, as well as comparing energies, partition functions, contribution to the population, etc. Starting geometries are tweaked to try and locate higher-energy conformers.
Note that it is intrinsically impossible to know if one has all conformers, and only valid conformers. This depends on the methodology used, and one can only strive to be as complete as possible for the chosen methodology. The problem of sampling the conformer space, and the need for improved techniques to handle large systems, is discussed in more detail in the supporting information in Novelli et al. 2020 (cited).

10) Line 340: comapred is a typo.

corrected

11) While I don't dispute that the SAR performs well, I wonder slightly at how impressive the factor of 2 agreement quoted in the abstract is, given that the total number of experimental data points included by the authors appears to be 22, and the SAR contains a fairly large number of parameters. (I understand the authors do not actually claim it is particularly impressive on its own, I would just like to hear their thoughts on this.)

For full-MC-TST calculations at an adequate level of theory (CCSD(T) on a geometry from a modern DFT functional, both at least with an aug-cc-pVTZ basis set), the accuracy is typically a factor of 2 to 3; this was verified in all the comparisons made with experiments and for all the reactions studied by us so far (some better, some worse).
A SAR is by definition less accurate because each SAR class lumps in many reactions. Furthermore, for some of the reaction classes in the current SAR, there are no calculations, or only low-level calculations available. Also, e.g. stereoisomers often have rate coefficients differing by an order of magnitude, such that uncertainties smaller than a factor 3-4 for all cases are unattainable except for extremely detailed SARs. The average factor 2 agreement with the experiments in this paper is thus at least partially fortuitous. The one example with a large deviation illustrates this well: it is based on the lowest-level data only, and this immediately deteriorates the performance of the SAR. Still, this suggests easy improvements by additional calculations. The range of spectator substitutions in the comparison is fairly wide, but the good agreement here means mostly that substituents not located on the reactive sites (H-atom, OO radical) are not affecting the rate all that much and need not be parameterized in the SAR. The factor 2 found is encouraging, but the sample set is too small to really validate the SAR. The theoretical data, with an order of magnitude scatter for full-MC-TST data on multi-functionalized species is probably a more accurate assessment of the expected overall accuracy.

To further illustrate the reliability of the SAR derivation procedure, the supporting information now includes a discussion of the transferability of the temperature dependence across reactions, illustrated with a set of graphs.

12) Do the authors have some guidelines on what to do for more complex systems not covered directly by the SAR? Obviously, the SAR prediction is a very good starting point and sanity check - if it's very low, then likely the H-shift will not be relevant anyway, but what about the cases where the SAR prediction is fairly high, and we need a more precise value? Would the next step be to find the closest reference system covered by the SAR, and attempt to apply the authors rel-MC-TST approach? (If yes, then it would DEFINITELY be good if the authors uploaded also their conformer data - as said I'm not sure from a quick glance if that's already included in the supplementary material)

We have no recommendations for molecules not covered by the SAR. The current SAR is a good start, but needs a lot more work before it is universally applicable with good accuracy. Still, that is a mammoth job, and should not stop us now from starting with a SAR, lest the perfect becomes the enemy of the good.
If a more accurate value is needed than available in the SAR, one could try measuring it, or do explicit rate calculations. Rel-MC-TST might be an option; all conformers are in the supporting information, or in the SI of earlier publications. Other publications likewise have made their conformers available.
However, depending on how much "more complex" the system is, one should consider that relative methods may not be the optimal choice (addressed in more detail in an earlier response).

[revised manuscript text omitted]